# Diet and nutrition status of adult multidrug-resistant tuberculosis cases, household controls, and community controls in Mumbai, India

Laxmi Govekar[1‡], Sabri Bromage[2,3‡*], Uttara Partap[4], Anupam Shukla[1], Prachi Dev[1], Yatin Dholakia[1], Wafaie W. Fawzi[3,4,5‡], Nerges Mistry[1‡]

1 Department of Tuberculosis Research, Foundation for Medical Research, Mumbai, India, 2 Community Nutrition Unit, Institute of Nutrition, Mahidol University, Phutthamonthon, Thailand, 3 Department of Nutrition, Harvard T.H. Chan School of Public Health, Boston, Massachusetts, United States of America, 4 Department of Global Health and Population, Harvard T.H. Chan School of Public Health, Boston, Massachusetts, United States of America, 5 Department of Epidemiology, Harvard T.H. Chan School of Public Health, Boston, Massachusetts, United States of America

‡ LG and SB authors are co-first author on this work. WWF and NM share co-senior author authorship on this work.

* sabri.bro@mahido.ac.th

## Abstract

India accounts for the largest national proportion of global multi-drug resistant (MDR-TB) cases and TB mortality. However, evidence on the role of diet and nutrition in MDR-TB infection remains limited. This study aimed to multifacetedly evaluate and compare diet and nutrition status of MDR-TB cases and controls in high TB-burden slum areas of Mumbai. We recruited 352 pulmonary MDR-TB cases receiving domiciliary treatment, household controls, and age-, sex-, and area-matched community controls 18–60 years of age. Participants were assessed for habitual food and nutrient intake using a validated semi-quantitative food-frequency questionnaire, other food consumption-related habits, diet quality metrics, anthropometry, biochemical measurements, and diet-related non-communicable diseases. Measures of diet and nutrition status were compared within and between study arms using hypothesis tests and multiple regression. The prevalence of dietary adequacy was < 50% for 18 of 24 assessed nutrients among cases and 12/24 nutrients among controls. Compared to both household and community controls, cases had significantly (p < 0.05) higher prevalence of underweight (66% vs. 23% and 15%, respectively), anemia (22% vs. 9% and 10%), and diabetes (18% vs. 4% and 5%); lower consumption of major healthy food groups including non-tuberous vegetables, deep orange vegetables, legumes, whole grains, and nuts and seeds; lower Global Diet Quality Score (GDQS); and higher prevalence of nutrient inadequacies including protein, thiamine, folate, and vitamins A, C, and E. Women had significantly poorer adequacy of most nutrients than men in all three study arms, and intake of most nutrients declined with

**Data availability statement:** All underlying data analyzed to produce the findings presented in this study have been de-identified and permanently uploaded to the following publicly accessible repository: https://doi.org/10.6084/m9.figshare.30000451.v1.

**Funding:** Financial support for this study was provided by Harvard Medical School Center for Global Health Delivery–Dubai (#VitD-027562-29356. SB acknowledges support from the National Institutes of Health Fogarty International Center (#5D43TW010543-05) and National Institute of Diabetes and Digestive and Kidney Diseases (#2T32DK007703-26). The funders played no role in the design, data collection, data analysis, or reporting of this study.

**Competing interests:** The authors have declared that no competing interests exist.

**Abbreviations:** 25(OH)D, 25-hydroxyvitamin D; 95%CI, 95% confidence interval; ATE, alpha-tocopherol equivalents BMI: body-mass index; DHS, demographic and health surveys; DS, drug-sensitive; EAR, estimated average requirement; FFQ, food frequency questionnaire; GDQS, global diet quality score; GDQS+, GDQS positive sub-metric; GDQS-, GDQS negative sub-metric; HFI, household food insecurity; HH, household; HIV, human immunodeficiency virus; aHR, adjusted hazard ratio; IFCT, Indian food composition table; IMS, India migration study; IQR, inter-quartile range; IMS, India migration study; LC-MS, liquid chromatography-mass spectrometry; MDR, multi-drug resistant; MUAC, mid-upper arm circumference; NFHS-5, national family health survey; NP WRA, non-pregnant women of reproductive age; NRIs, nutrient requirements for Indians; NTBPS, national TB prevalence survey; OR, odds ratio; QA/QC, quality assurance/quality control; RAE, retinol activity equivalents; RATIONS, reducing activation of tuberculosis by improvement of nutritional status; SD, standard deviation; SSBs, sugar-sweetened beverages; TB, tuberculosis; TEI, total energy intake; WC, waist circumference; USDA, United States Department of Agriculture; WHO, World Health Organization.

asset index and age in models adjusted for age, sex, study arm, and asset index. Results indicate an urgent need to improve diet and nutrition in Mumbai slum dwellers – particularly among the MDR-TB-infected population, women, the elderly, and the poorest households – and highlights the potentially key role of nutrition interventions in reducing MDR-TB burden in urban India.

## Introduction

Tuberculosis (TB) remains an important public health problem, with over 8 million people diagnosed with the disease in 2023. Considered to be the leading cause of death from an infectious agent, TB has a high number of incident cases particularly in India [1]. India has made steady progress in mitigating the national burden of TB since the early 1990s, thanks to improvements in successive TB elimination programs, health service provision, and poverty reduction. However, the country yet accounts for the largest national proportion of global TB cases (27%), multi-drug resistant (MDR-TB) cases (27%), undiagnosed TB cases (18%), and TB mortality (26%) [2,3]. The 2019–2021 National TB Prevalence Survey (NTBPS) found that 22.6% (95%CI: 19.4%-25.8%) of Indians ≥15 years of age are infected with TB as measured using the QuantiFERON-TB Gold Plus assay [4], among which approximately 3% and 13% of new and previously treated cases have MDR-TB or rifampicin-resistant TB [3]. Two recent meta-analyses of 79 and 32 surveys from 1989-2016 estimated MDR-TB prevalence at 27% and 35%, respectively, among treated cases [5,6]. Significant gaps in treatment coverage pose a major obstacle to India's TB elimination strategy [3]. The burden of TB and MDR-TB is especially severe in urban India. NTBPS found an adjusted risk ratio of 1.59 (95%CI: 1.04-2.44) for TB infection among those residing in urban areas as compared to rural ones [4]. Urban Indians are at higher risk of TB infection and progression to active disease due to overcrowding, internal migration, poor sanitation, indoor and outdoor air pollution, and high prevalence of relevant comorbidities such as acute respiratory infections, HIV, hypertension, and type 2 diabetes [4,7–9]. The prevalence of TB in slum dwellers of Mumbai has been observed to be as high as 5,150 per 100,000 [7] and 24%–30% of new cases and 11%–67% of treated cases in Mumbai have MDR-TB [9]. Compared with drug-sensitive (DS) TB, MDR-TB in India is associated with markedly higher treatment costs, rates of treatment failure, and case fatality rates [10,11].

Malnutrition, encompassing both under- and overnutrition, is also a major risk factor for TB. The 2019–2021 National Family Health Survey (NFHS-5) found that 18.7% of Indian women and 15.6% of men aged 15–49 are underweight, 24.0% of women and 23.7% of men are overweight, and 57.0% of women and 25.1% of men are anemic [12]. Underweight, protein-energy malnutrition, and micronutrient deficiencies impair cell-mediated immunity and lead to delayed sputum conversion, more severe clinical presentation of TB, malabsorption of anti-tubercular medications, drug-induced hepatotoxicity, relapse after cure, and death [13–18]. TB infection itself exacerbates undernutrition by increasing nutritional requirements, stimulating catabolic processes, and decreasing appetite [15–18]. Nutrition-related cardiometabolic

risks including obesity and impaired fasting glucose also contribute to TB infection, progression, and severity by disrupting immune function through chronic inflammation, vascular dysfunction, and dysregulation of nutrient metabolism, and by altering pharmacokinetics during TB treatment [19,20].

Despite evidence as to the role of nutrition in TB infection and disease, there remains limited understanding of the role of modifiable nutritional factors such as diet quality and nutrient adequacy, particularly as these factors pertain to risk of MDR-TB. Understanding the links between diet and disease is important for informing nutritional interventions to control TB, especially in high-TB burden populations that are highly malnourished and among which widespread drug resistance calls for a more holistic approach in both clinical and public health settings. To investigate these links, this study sought to multi-facetedly evaluate and compare the diets of MDR-TB cases, household controls, and community controls in a slum-dwelling population of Mumbai, and establish a foundation for research and programmatic efforts to combat TB in urban India.

## Methods

### Study population

This study involved secondary analysis of a case-control study [21,22] conducted among 90 pulmonary MDR-TB cases receiving outpatient (domiciliary) treatment from two public hospitals in the M-East, M-West, and H-East wards (local administrative units [23]); 180 household controls; and 82 community controls. Non-household controls were recruited from non-respiratory departments at the hospitals where the cases were identified, and were matched with cases by age, sex, and ward of residence. Cases were required to have resided in the M-East, M-West, or H-East wards for ≥6 months, initiated domiciliary treatment for MDR-TB < 1 month previously, no history of MDR-TB in the past 2 years, and ≥2 eligible household controls. Household controls must have resided with their index case for ≥1 year prior to the case's diagnosis and have had no active TB symptoms at screening, and community controls must have had neither active TB symptoms at screening nor a family member with a history of TB in the past 2 years. All participants were required to be 18–60 years of age and non-pregnant.

Participants were recruited and subjected to assessments from January 28, 2020 to December 17, 2020. All participants provided written informed consent prior to enrolment. The study protocol was approved by the Harvard T.H. Chan School of Public Health Institutional Review Board (#IRB19–0237), Foundation for Medical Research Institutional Research Ethics Committee (#FMR/IREC/TB/01/2019), and Health Ministry's Screening Committee of the Indian Council for Medical Research (# 2019–7974).

### Data collection procedures

Following enrolment, participants were assessed for demographic, socioeconomic (including household assets), and clinical characteristics; details on diet in the past year; and individual and household food consumption-related habits. As this was a case-control study, data were collected only once from each participant.

Dietary data collection details. During the study visit, habitual consumption of foods and beverages over the past year was recorded using a 195-item semi-quantitative food-frequency questionnaire (FFQ), previously validated for use in urban and rural India as part of the India Migration Study (IMS) [24], and adapted to include 11 new foods commonly consumed in Mumbai. The IMS FFQ used in this study has demonstrated acceptable validity and reproducibility across urban and rural Indian populations [24]; FFQs typically exhibit error margins of approximately 20–40% in estimating absolute nutrient intake compared to reference methods, yet remain reliable tools for ranking intakes in epidemiological research [25]. The FFQ and other questionnaires were administered by trained field workers using the SurveyCTO platform. For each food, frequency of consumption per year, month, week, or day was recorded depending on the most convenient frequency basis for the participant, and the number of portions typically consumed per meal was also recorded. For 27 fruits and vegetables, participants were also asked whether these foods were only consumed when they were seasonally available. For foods typically served using utensils, typical portion sizes were assessed using physical

models of five standard serving sizes (teaspoon, tablespoon, ladle, bowl, and glass) displayed to participants during the dietary assessment.

Food consumption-related habits. We included a set of questions regarding individual and household food consumption-related habits. This included, but was not limited to, questions on household use of different types of fats, oils, thickening agents, and milks, fasting and adherence to special diets, and use of alcohol and supplements.

Clinical data collection details. Clinical histories were obtained from medical records (cases) and self-reports (controls). Participant height, weight, waist circumference (WC), and mid-upper arm circumference (MUAC) were measured using portable stadiometer, scale, and anthropometric tape. Hemoglobin and random blood glucose measurements were collected using Mission ULTRA hemoglobinometer (Acon Labs, San Diego, CA, USA) and Contour TS glucometer (Ascensia Diabetes Care UK Limited, Newbury, UK). Blood was collected by a certified phlebotomist to measure serum total 25-hydroxyvitamin D (25(OH)D) at the Metropolis Healthcare laboratory in Mumbai using LC-MS with QA/QC measures set by the India National Accreditation Board for Testing and Calibration Laboratories (Gurgaon, India). Among controls, we also assessed TB infection using QFT-G IGRA.

## Dietary assessment

Using the available dietary (FFQ) data, we calculated food and food group intake, nutrient intake and adequacy, and diet metric scores. Calculations were done using Microsoft Excel [26]. Details are noted below.

The daily mass of each food consumed in g/day was calculated by multiplying its consumption frequency, number of portions, typical portion size, and grams per serving based on food-specific density estimates derived from the IMS (some of these estimates were adjusted to more accurately reflect the actual size of utensils used in the current study, some of which differed from those in the IMS given regional variation in culinary practices). For fruits and vegetables reported consumed in season only, the consumed mass was further adjusted depending on the number of months per year during which the food was seasonally available using data from the India Agricultural Produce Market Committee [27].

Food consumption data were used to calculate consumption of key food groups in g/day and g/2000 kcal/day (to aid comparisons across subgroups, with the cutoff of 2000 kcal/day based on national estimates of mean daily caloric intake [28]), and consumption of a separate set of "dish-based" food groups (food groups that are distinguished by local culinary practices instead of inherent nutritional significance). Food consumption was also used to calculate the Global Diet Quality Score (GDQS, a food-based metric of undernutrition and diet-related non-communicable disease risk that awards points for higher consumption of healthy food groups and lower consumption of healthy food groups; range: 0–49), GDQS positive and negative sub-metrics (GDQS+ and GDQS-, computed using only healthy and unhealthy GDQS food groups, respectively, higher scores of which indicate higher consumption of healthy foods and lower consumption of unhealthy foods, respectively; range: 0–32 and 0–17, respectively), by assigning GDQS food groups at the level of individual ingredients (parsed using recipe data from IMS) and applying published scoring guidance [29].

Nutrient intake was calculated using IMS recipe and food composition data, the latter of which were compiled largely from the 2017 Indian Food Composition Table (IFCT) [30] and supplemented using data from the USDA for vitamin D [31]. Given anomalously high [32] vitamin D concentrations for plant-source foods in the IFCT, these were replaced with values from the USDA in the current study. For nutrients except iron, dietary adequacy was classified using the estimated average requirement (EAR) cut-point method [33], drawing EARs from the 2020 Nutrient Requirements for Indians (NRIs) [34]. EARs were assigned based on age, sex, and physical activity level (sedentary, moderate, or heavy) classified based on participants' occupations [35]. For nutrients without established EARs in the NRIs, adequacy was classified using Recommended Intake Levels (Cu, K, Mn, Na, P, Se) if available in the NRIs or otherwise using Adequate Intake Levels in the US Dietary Reference Intakes (fiber and pantothenic acid) [33]. The prevalence of iron adequacy was calculated using the full probability approach, assuming moderate usage of oral contraceptives [36] and bioavailabilities of 5% and 8% for men and women, respectively, as advised in the NRIs.

## Generation of clinical variables

Body mass index (BMI) was calculated and classified as underweight (<18.5), healthy (18.5–22.9), overweight (23–24.9), or obese (≥25) according to consensus guidelines for Asian Indians [37]. Waist circumference was defined as >102 cm for men and >88 cm for women [38], low MUAC as <25.5 cm for men and 24.5 cm for women [39] and anemia as hemoglobin <13 g/dl for men and <12 g/dl for women [40]. Vitamin D status was categorized based on serum 25(OH)D as deficient (<10 ng/ml), insufficient (10−<20), adequate (20−<30), or optimal (≥30 ng/mL) [41].

We used asset index as a key indicator of socioeconomic status (SES). There are multiple measures of SES, including education, employment, income, expenditure and assets [42]. Ownership of specific assets is considered a generally reliable and useful indicator of longer-term socioeconomic status. We used principal components analysis (PCA) to construct an asset index based on socioeconomic data on the availability of a range of assets in the household (e.g., bicycle, television, furniture, pressure cooker, etc.). PCA is a statistical data reduction technique that encompasses a linear combination of multiple correlated variables, creating a set of uncorrelated components [42]. We used the first component generated from the PCA as the asset index score, and categorized it into quartiles. Demographic and Health Survey (DHS) methodology [43] was adapted to compute the asset index using the *prcomp* function [44] in R.

## Statistical analysis

Within nine subgroups of the study population – consisting of the three study arms (cases, household controls, and community controls) and the six male and female subsets of each study arm – descriptive statistics were computed to describe distributions of food group and nutrient intake, prevalence of nutrient adequacy, diet metric scores, measures of nutrition status, and individual and household food consumption-related habits. Significance of pairwise differences in means and prevalences between study arms and between male and female subsets of each study arm was evaluated using t-tests for continuous variables and Fisher's exact tests for categorical variables.

Multiple linear regression models were run to estimate marginal mean intake of key food groups and nutrients, and diet metric scores within age groups (<25, 25-<40, and ≥40 years) and quartiles of asset index, adjusted for one another (age category and asset quartile), sex, and study arm; for each model, tests for linear trend were conducted to evaluate trends in each outcome across age categories and asset quartiles. Each model was repeated after incorporating an interaction term between study arm and either sex, age category, or asset quartile, in order to examine potential effect modification by study arm. Diet metrics were analyzed as both crude and energy-adjusted variables (adjusted using the residual method). Energy adjustment was done in order to examine trends in nutrient intake across subgroups that were independent of total energy intake [45].

Statistical analyses were performed in R version 4.0.4 (R Foundation for Statistical computing, Vienna, Austria) and Stata 16 (StataCorp,Texas, USA).

## Results

### Participant characteristics

Approximately 62%, 27% and 11% of participants were sampled from M: East, M: West and H: East wards respectively. The mean age of participants was 33.8 ± 12.0 years. Household controls were significantly (p < 0.05) older (36.8 years) than both cases (29.8 years) and community controls (31.6 years). Women comprised 54% of participants. Over 60% of participants had a secondary education, and about 70% reported having 3 or more household members sleeping per room. About one third of household (36%) and community controls (32%) had a positive QFT-G IGRA test. Detailed descriptive characteristics can be found in Table 1 of Shukla 2022, et al. [22].

**Table 1. Mean (SD) percentage of total energy intake contributed by different category of consumed dishes by population subgroup.**

| Category | Cases, N = 90 | HH controls, N = 180 | Community controls, N = 82 | Male cases, N = 42 | Female cases, N = 48 | Male HH controls, N = 84 | Female HH controls, N = 96 | Male community controls, N = 37 | Female community controls, N = 45 |
|---|---|---|---|---|---|---|---|---|---|
| Rice dishes | 31.8 (13.6) | 33.3 (14.6)¥ | 28.7 (12.0)¥ | 32.1 (13.8) | 31.5 (13.7) | 33.8 (15.1) | 32.8 (14.1) | 28.8 (13.1) | 28.6 (11.2) |
| Sweets | 13.2 (9.1)* | 9.6 (7.8)*¥ | 12.3 (8.0)¥ | 12.9 (9.7) | 13.5 (8.7) | 9.0 (7.3) | 10.0 (8.3) | 11.2 (7.8) | 13.3 (8.1) |
| Breads and rotis | 6.8 (7.8)† | 9.3 (11.2) | 11.6 (11.5)† | 8.2 (8.7) | 5.6 (6.7) | 8.7 (11.1) | 9.8 (11.3) | 9.4 (9.2) | 13.5 (12.9) |
| Meat, poultry, and eggs | 7.3 (5.9) | 7.6 (6.7) | 7.3 (6.0) | 8.1 (6.4) | 6.5 (5.4) | 9.2 (7.2)§ | 6.3 (5.9)§ | 8.3 (6.1) | 6.4 (5.8) |
| Vegetarian curries and soups | 7.0 (4.9) | 8.1 (5.9) | 6.7 (3.8) | 6.4 (4.5) | 7.5 (5.3) | 8.4 (5.7) | 7.8 (6.0) | 6.9 (3.9) | 6.6 (3.7) |
| Non-alcoholic drinks | 7.8 (6.2)† | 7.9 (6.6)¥ | 5.9 (5.4)†¥ | 7.0 (5.5) | 8.5 (6.8) | 7.0 (5.4) | 8.6 (7.4) | 5.2 (6.0) | 6.4 (4.9) |
| Fritters | 5.8 (6.0)* | 4.3 (5.3)*¥ | 6.2 (6.3)¥ | 3.4 (3.4)‡ | 8.0 (6.9)‡ | 4.2 (5.9) | 4.4 (4.7) | 5.9 (6.7) | 6.4 (6.1) |
| Fish and seafood | 5.1 (8.5) | 4.6 (7.8) | 4.2 (7.0) | 7.2 (8.9)‡ | 3.3 (7.8)‡ | 4.3 (7.8) | 4.8 (7.9) | 4.6 (8.2) | 3.9 (5.8) |
| Fruits | 4.3 (5.9) | 4.8 (5.4) | 4.4 (6.4) | 3.3 (3.8) | 5.1 (7.1) | 4.5 (4.6) | 5.1 (6.1) | 5.0 (8.8) | 3.9 (3.4) |
| Vegetables | 2.8 (1.8) | 3.1 (2.7) | 2.7 (2.0) | 2.4 (1.6) | 3.1 (2.0) | 3.0 (2.5) | 3.1 (2.8) | 3.0 (1.8) | 2.5 (2.1) |
| Milk and dairy | 2.5 (3.4) | 2.2 (3.6) | 2.3 (3.6) | 2.6 (3.3) | 2.4 (3.5) | 2.3 (3.4) | 2.1 (3.7) | 2.7 (4.6) | 1.9 (2.4) |
| Other grain-based dishes | 1.7 (2.4)† | 1.8 (2.5)¥ | 2.6 (2.3)†¥ | 2.0 (3.0) | 1.4 (1.6) | 2.0 (2.8) | 1.7 (2.2) | 2.8 (2.4) | 2.5 (2.2) |
| Papad, pickles, and chutney | 1.4 (1.4)† | 1.6 (1.6)¥ | 2.3 (2.0)†¥ | 1.4 (1.6) | 1.5 (1.2) | 1.5 (1.5) | 1.8 (1.7) | 2.1 (2.4) | 2.5 (1.6) |
| Other dishes | 1.0 (2.6) | 0.7 (1.3)¥ | 1.0 (1.3)¥ | 0.5 (0.5) | 1.5 (3.5) | 0.6 (1.0) | 0.8 (1.6) | 1.4 (1.8)# | 0.8 (0.7)# |
| Roots, tubers, and plantains | 0.6 (0.6)* | 0.9 (1.0)* | 1.0 (1.9) | 0.5 (0.4)‡ | 0.8 (0.6)‡ | 0.9 (1.2) | 0.8 (0.9) | 1.2 (2.6) | 0.9 (1.1) |
| Alcoholic drinks | 0.9 (3.2)* | 0.3 (1.2)*¥ | 0.7 (2.5)¥ | 1.8 (4.5)‡ | 0.0 (0.0)‡ | 0.5 (1.8)§ | 0.0 (0.1)§ | 1.6 (3.6)# | 0.0 (0.0)# |

Footnote: Shading is proportional to mean percentage contribution (white: 0.0%, darkest blue: 33.8%). p for difference <0.05 (t-test): * cases vs. HH controls, † cases vs. community controls, ¥ HH vs. community controls, ‡ male vs. female cases, § male vs. female HH controls, # male vs. female community controls. Abbreviation: HH, household.

## Food consumption patterns across cases and controls

We examined food consumption across case and control groups, including energy intake from different dishes (Table 1), consumption of key food groups (Table 2), diet quality (Table 3), and other food habits including fasting and household meal preparation (S1 Table).

Percentage of energy intake contributed by different dishes. Over half of total energy intake was contributed by rice-based dishes (contributing 31.8%, 33.3%, and 28.7% in cases, household and community controls, respectively), sweets (13.2%, 9.6%, and 12.3%, respectively), and breads and rotis (6.8%, 9.3%, and 11.6%, respectively) (Table 1).

Consumption and consumption density of key food groups. Food groups comprising the bulk of cases' diets included flour products (mean intake: 540.1 g/day), non-tuberous vegetables (247.4 g/day, of which 77.7 g/day were legumes), milk and dairy products (220.7 g/day), and fruits (109.2 g/day) (Table 2). Cases had significantly (p < 0.05) lower mean intake than household controls for non-tuberous vegetables (274.4 vs. 299.1 g/day), deep orange vegetables (4.2 vs. 7.1 g/day), legumes (77.7 vs. 100.1 g/day), and total (540.1 vs. 636.6 g/day) as well as whole (5.9 vs. 11.4 g/day) grains and flour products; and significantly lower mean intake than community controls for deep orange vegetables (4.2 vs. 9.0 g/day), nuts and seeds (5.8 vs. 8.8 g/day), and whole grains and flour products (5.9 vs. 13.2 g/day) (Table 2). Community controls had significantly higher intake than household controls for poultry (45.2 vs. 31.1 g/day), hydrogenated oils (1.8 vs. 0.7 g/day), and purchased deep fried foods (54.3 vs. 41.3 g/day) (Table 2).

Men had significantly higher mean consumption than women for non-tuberous vegetables and liquid oils within all three study arms (Table 2). Men also had significantly higher intake of refined grains and flour products, and eggs within both case and household control arms; legumes and poultry within both household and community controls; and dark green leafy vegetables, other vegetables, total animal-source foods, milk and dairy products, and fish within cases (Table 2). In

**Table 2. Mean (SD) consumption (g/day) and consumption density (g/2000 kcal/day) of key food groups by population subgroup.**

| Food group | Cases, N=90 | HH controls, N=180 | Community controls, N=82 | Male cases, N=42 | Female cases, N=48 | Male HH controls, N=84 | Female HH controls, N=96 | Male community controls, N=37 | Female community controls, N=45 |
|---|---|---|---|---|---|---|---|---|---|
| *Consumption (g/day)* | | | | | | | | | |
| Fruits | 109.2 (132.6) | 134.5 (160.0) | 149.1 (201.9) | 115.9 (139.3) | 103.4 (127.6) | 136.7 (171.0) | 132.5 (150.5) | 180.8 (275.4) | 123.0 (107.0) |
| Citrus fruits | 21.0 (42.7) | 21.9 (33.3) | 30.8 (61.6) | 20.0 (30.6) | 21.9 (51.2) | 18.8 (27.7) | 24.6 (37.5) | 38.6 (87.5) | 24.3 (24.9) |
| Deep orange fruits | 15.5 (31.7) | 28.0 (76.1) | 22.3 (26.1) | 11.7 (20.3) | 18.9 (39.0) | 22.1 (64.4) | 33.2 (85.1) | 22.2 (28.0) | 22.3 (24.7) |
| Other fruits | 72.7 (107.8) | 84.6 (95.5) | 96.0 (187.1) | 84.2 (124.2) | 62.6 (91.3) | 95.9 (113.4) | 74.7 (75.8) | 119.9 (265.0) | 76.4 (78.2) |
| Non-tuberous vegetables | 247.4 (176.7)* | 299.1 (185.1)* | 292.3 (180.0) | 288.4 (169.5)‡ | 211.6 (176.8)‡ | 331.4 (191.4)‡ | 270.9 (175.5)§ | 339.4 (187.0)# | 253.5 (166.2)# |
| Cruciferous vegetables | 10.7 (12.4) | 12.1 (11.6) | 11.7 (9.6) | 11.6 (13.6) | 9.9 (11.4) | 11.8 (10.4) | 12.3 (12.6) | 13.2 (10.8) | 10.5 (8.4) |
| Dark green leafy vegetables | 17.5 (23.1) | 17.8 (17.5) | 19.5 (19.1) | 24.7 (30.3)‡ | 11.3 (11.0)‡ | 20.0 (21.0) | 15.9 (13.6) | 21.8 (19.3) | 17.6 (18.9) |
| Deep orange vegetables | 4.2 (4.9)*† | 7.1 (10.8)* | 9.0 (19.4)† | 5.1 (6.1) | 3.4 (3.5) | 8.8 (14.0) | 5.7 (6.8) | 12.3 (27.9) | 6.3 (6.5) |
| Legumes | 77.7 (60.1)* | 100.1 (74.4)* | 90.9 (49.2) | 86.5 (55.7) | 70.0 (63.3) | 113.7 (82.1)§ | 88.2 (65.1)§ | 105.4 (50.1)# | 78.9 (45.5)# |
| Other vegetables | 137.3 (101.1) | 162.0 (100.6) | 161.2 (110.3) | 160.4 (98.5)‡ | 117.0 (99.9)‡ | 177.2 (102.1) | 148.8 (97.9) | 186.7 (115.0) | 140.2 (102.8) |
| Nuts and seeds | 5.8 (5.2)† | 7.6 (8.3) | 8.8 (6.0)† | 6.2 (4.9) | 5.5 (5.5) | 7.7 (7.5) | 7.5 (9.0) | 10.1 (7.0) | 7.7 (4.9) |
| White roots and tubers | 17.3 (19.9) | 16.8 (16.9) | 20.3 (21.1) | 16.5 (15.9) | 18.0 (23.0) | 17.8 (19.6) | 15.9 (14.1) | 20.9 (23.1) | 19.8 (19.4) |
| Grains and flour products | 540.1 (353.7)* | 636.6 (391.7)* | 612.4 (391.3) | 661.9 (369.4)‡ | 433.5 (304.9)‡ | 712.4 (452.4)§ | 570.3 (317.5)§ | 691.7 (454.6) | 547.1 (321.2) |
| Refined grains and flour products | 534.2 (353.4) | 625.2 (385.5) | 599.2 (383.9) | 654.2 (372.3)‡ | 429.3 (302.5)‡ | 701.4 (447.0)§ | 558.6 (309.6)§ | 675.2 (439.4) | 536.7 (323.3) |
| Whole grains and flour products | 5.9 (12.8)*† | 11.4 (22.0)* | 13.2 (27.8)† | 7.7 (16.0) | 4.3 (8.9) | 11.1 (22.0) | 11.7 (22.2) | 16.5 (38.7) | 10.5 (13.2) |
| Animal-source foods | 321.6 (233.6) | 366.4 (315.7) | 330.7 (217.0) | 401.1 (270.3)‡ | 252.0 (170.2)‡ | 395.8 (319.5) | 340.8 (311.6) | 374.9 (249.4) | 294.4 (181.1) |
| Milk and dairy products | 220.7 (140.1) | 264.7 (255.4) | 228.2 (165.1) | 262.6 (162.1)‡ | 184.1 (106.4)‡ | 275.5 (253.7) | 255.3 (257.9) | 243.1 (183.8) | 216.0 (149.1) |
| Eggs | 22.2 (42.3) | 22.3 (34.9) | 21.6 (30.7) | 33.0 (54.1)‡ | 12.8 (25.1)‡ | 31.8 (41.9)§ | 14.1 (24.6)§ | 26.4 (30.8) | 17.7 (30.4) |
| Red meat | 9.6 (14.7) | 8.8 (16.7) | 6.8 (16.9) | 11.9 (16.4) | 7.6 (12.9) | 10.2 (19.1) | 7.6 (14.4) | 8.2 (23.5) | 5.6 (8.7) |
| Processed meat | 0.0 (0.0) | 0.0 (0.2) | 0.0 (0.0) | 0.0 (0.0) | 0.0 (0.0) | 0.0 (0.0) | 0.0 (0.2) | 0.0 (0.0) | 0.0 (0.0) |
| Poultry | 29.0 (44.3) | 31.1 (36.2)* | 45.2 (73.4)* | 34.0 (43.2) | 24.6 (45.2) | 37.8 (34.0)§ | 25.3 (37.2)§ | 63.9 (101.1)# | 29.7 (31.9)# |
| Fish | 40.0 (88.1) | 39.5 (100.0) | 28.9 (50.3) | 59.6 (103.5)‡ | 22.8 (68.7)‡ | 40.6 (100.7) | 38.5 (100.0) | 33.3 (58.9) | 25.3 (42.3) |
| Juice and SSBs | 31.0 (90.6) | 25.7 (56.5) | 33.2 (71.6) | 48.4 (127.4) | 15.7 (29.3) | 27.2 (60.4) | 24.4 (53.1) | 46.4 (99.4) | 22.3 (33.1) |
| Juice | 10.9 (26.1) | 11.9 (30.6) | 19.5 (62.5) | 13.4 (30.1) | 8.6 (22.0) | 10.7 (25.1) | 12.9 (34.8) | 28.2 (89.1) | 12.4 (24.0) |
| SSBs | 20.1 (80.7) | 13.8 (43.8) | 13.6 (30.5) | 35.0 (116.4) | 7.1 (11.3) | 16.5 (50.9) | 11.5 (36.5) | 18.3 (38.5) | 9.9 (21.6) |
| Sweets and caloric sweeteners | 49.4 (44.2) | 46.4 (47.5) | 49.6 (35.3) | 57.1 (52.1) | 42.7 (35.2) | 43.5 (42.7) | 49.0 (51.4) | 47.6 (36.9) | 51.2 (34.2) |
| Oils | 42.6 (31.6) | 50.3 (36.5) | 51.8 (29.8) | 49.8 (32.4)‡ | 36.2 (29.8)‡ | 56.4 (38.5)§ | 45.0 (33.9)§ | 60.3 (32.2)# | 44.9 (26.1)# |
| Liquid oils | 41.9 (30.9) | 49.6 (35.9) | 50.0 (27.4) | 49.1 (32.0)‡ | 35.5 (28.8)‡ | 55.6 (38.1)§ | 44.3 (33.2)§ | 57.3 (28.3)# | 44.0 (25.5)# |
| Hydrogenated oils | 0.7 (1.7) | 0.7 (1.8)* | 1.8 (5.7)* | 0.7 (1.1) | 0.7 (2.1) | 0.8 (1.1) | 0.6 (2.3) | 3.0 (8.2) | 0.9 (1.4) |

*(Continued)*

Table 2. (Continued)

| Food group | Cases, N=90 | HH controls, N=180 | Community controls, N=82 | Male cases, N=42 | Female cases, N=48 | Male HH controls, N=84 | Female HH controls, N=96 | Male community controls, N=37 | Female community controls, N=45 |
|---|---|---|---|---|---|---|---|---|---|
| Purchased deep fried foods | 44.6 (46.5) | 41.3 (47.3)* | 54.3 (49.9)* | 45.4 (46.4) | 43.9 (47.1) | 47.4 (53.9) | 36.0 (40.2) | 60.0 (59.9) | 49.5 (39.9) |
| *Consumption (g/2000 kcal/day)* | | | | | | | | | |
| Fruits | 106.6 (101.8) | 128.4 (120.6) | 128.2 (135.8) | 93.1 (86.6) | 118.3 (113.1) | 117.1 (93.6) | 138.2 (139.9) | 135.7 (168.5) | 122.0 (103.0) |
| Citrus fruits | 17.9 (29.3) | 20.4 (30.7) | 28.2 (57.5) | 14.9 (21.4) | 20.6 (34.8) | 15.7 (21.1) | 24.6 (36.8) | 31.5 (81.3) | 25.5 (25.6) |
| Deep orange fruits | 15.4 (24.9) | 24.1 (49.1) | 21.2 (27.9) | 10.0 (18.9) | 20.1 (28.6) | 17.8 (25.1) | 29.6 (62.7) | 19.2 (27.0) | 22.8 (28.8) |
| Other fruits | 73.2 (92.3) | 83.9 (91.3) | 78.8 (112.8) | 68.3 (80.1) | 77.6 (102.5) | 83.7 (79.0) | 84.1 (101.2) | 85.0 (149.4) | 73.8 (71.5) |
| Non-tuberous vegetables | 264.9 (112.6) | 289.1 (137.2) | 260.6 (102.8) | 251.0 (92.2) | 277.0 (127.6) | 297.4 (132.5) | 281.9 (141.6) | 268.5 (111.0) | 254.0 (96.4) |
| Cruciferous vegetables | 11.4 (10.3) | 11.9 (9.4) | 11.1 (7.8) | 10.3 (11.0) | 12.4 (9.7) | 11.0 (8.8) | 12.7 (9.8) | 11.2 (8.9) | 11.0 (6.9) |
| Dark green leafy vegetables | 16.9 (13.7) | 17.4 (13.7) | 16.5 (11.0) | 19.3 (15.9) | 14.8 (11.3) | 18.6 (15.0) | 16.4 (12.4) | 16.3 (9.9) | 16.6 (12.0) |
| Deep orange vegetables | 4.5 (6.6) | 6.3 (9.2) | 7.5 (13.7) | 4.0 (3.6) | 5.0 (8.4) | 7.2 (12.4) | 5.5 (4.9) | 8.8 (19.2) | 6.4 (6.5) |
| Legumes | 86.1 (43.9) | 96.5 (57.8) | 83.8 (39.9) | 79.4 (37.8) | 92.0 (48.3) | 101.0 (57.7) | 92.6 (58.0) | 86.5 (43.0) | 81.5 (37.6) |
| Other vegetables | 145.9 (62.9) | 157.0 (74.9) | 141.8 (54.9) | 138.1 (51.2) | 152.8 (71.4) | 159.7 (71.1) | 154.7 (78.4) | 145.8 (59.3) | 138.4 (51.5) |
| Nuts and seeds | 6.1 (5.7)† | 6.6 (5.4) | 8.0 (6.1)† | 5.0 (2.7) | 7.1 (7.3) | 6.2 (4.3) | 7.0 (6.3) | 8.1 (6.3) | 7.9 (6.0) |
| White roots and tubers | 18.6 (15.6) | 15.5 (12.1) | 17.4 (13.6) | 13.6 (10.1)‡ | 22.8 (18.1)‡ | 15.3 (14.0) | 15.6 (10.2) | 14.9 (13.0) | 19.4 (13.8) |
| Grains and flour products | 571.9 (234.3) | 606.0 (257.1)¥ | 529.7 (201.2)¥ | 586.2 (250.1) | 559.4 (221.6) | 603.1 (268.9) | 608.5 (247.8) | 519.2 (212.8) | 538.3 (193.1) |
| Refined grains and flour products | 566.6 (238.1) | 596.8 (258.3)¥ | 518.5 (199.3)¥ | 580.2 (254.4) | 554.7 (224.9) | 595.5 (270.3) | 598 (248.7) | 508.0 (205.5) | 527.0 (195.9) |
| Whole grains and flour products | 5.3 (10.8)† | 9.2 (17.3) | 11.2 (17.1)† | 6.0 (11.5) | 4.7 (10.2) | 7.6 (14.9) | 10.6 (19.1) | 11.2 (20.7) | 11.3 (13.8) |
| Animal-source foods | 360.5 (192.9)† | 350.4 (191.9)¥ | 287.9 (113.8)†¥ | 361.0 (189.4) | 360.1 (197.9) | 345.2 (173.7) | 355.0 (207.3) | 286.3 (117.1) | 289.2 (112.4) |
| Milk and dairy products | 273.9 (181.6)† | 265.6 (188.2)¥ | 204.1 (115.3)†¥ | 254.3 (156.8) | 291.1 (200.8) | 248.8 (162.6) | 280.3 (207.7) | 192.2 (120.8) | 213.9 (110.9) |
| Eggs | 21.5 (41.2) | 20.7 (35.7) | 20.7 (36.5) | 29.1 (52.4) | 14.8 (27.0) | 29.1 (46.9)§ | 13.4 (19.1)§ | 22.3 (27.0) | 19.5 (43.1) |
| Red meat | 9.5 (15.1)† | 8.8 (16.9) | 5.4 (8.7)† | 9.0 (11.1) | 10.0 (18.0) | 9.0 (16.5) | 8.6 (17.3) | 4.9 (9.6) | 5.8 (8.0) |
| Processed meat | 0.0 (0.0) | 0.0 (0.1) | 0.0 (0.0) | 0.0 (0.0) | 0.0 (0.0) | 0.0 (0.0) | 0.0 (0.1) | 0.0 (0.0) | 0.0 (0.0) |
| Poultry | 24.5 (23.3) | 27.5 (23.9) | 32.1 (27.9) | 25.0 (20.2) | 24.0 (25.9) | 32.1 (24.9)§ | 23.5 (22.4)§ | 39.1 (34.7)# | 26.4 (19.4)# |
| Fish | 31.1 (51.1) | 27.8 (47.2) | 25.5 (42.1) | 43.5 (53.5)‡ | 20.2 (46.9)‡ | 26.2 (47.2) | 29.2 (47.5) | 27.8 (49.7) | 23.6 (35.0) |
| Juice and SSBs | 24.2 (52.1) | 21.1 (39.6) | 27.6 (62.0) | 31.6 (72.8) | 17.7 (20.8) | 19.8 (33.5) | 22.2 (44.4) | 35.4 (86.4) | 21.2 (29.5) |
| Juice | 8.4 (17.6) | 11.0 (29.1) | 17.7 (60.1) | 9.5 (21.2) | 7.5 (13.8) | 8.8 (20.0) | 13.0 (35.2) | 24.4 (85.7) | 12.2 (23.9) |
| SSBs | 15.7 (46.2) | 10.1 (25.6) | 9.9 (18.9) | 22.1 (65.8) | 10.1 (13.6) | 11.0 (25.3) | 9.3 (26.0) | 11.0 (19.2) | 9.0 (18.9) |

*(Continued)*

**Table 2.** (Continued)

| Food group | Cases, N=90 | HH controls, N=180 | Community controls, N=82 | Male cases, N=42 | Female cases, N=48 | Male HH controls, N=84 | Female HH controls, N=96 | Male community controls, N=37 | Female community controls, N=45 |
|---|---|---|---|---|---|---|---|---|---|
| Sweets and caloric sweeteners | 52.8 (28.4)*† | 42.3 (24.9)* | 43.4 (22.5)† | 50.2 (30.6) | 55.2 (26.4) | 36.7 (19.9)§ | 47.3 (27.7)§ | 36.4 (19.6)# | 49.1 (23.2)# |
| Oils | 43.3 (14.2) | 45.3 (15.6) | 44.8 (11.7) | 41.6 (12.9) | 44.8 (15.3) | 46.9 (15.1) | 43.9 (16.0) | 46.3 (12.0) | 43.5 (11.5) |
| Liquid oils | 42.7 (14.2) | 44.7 (15.5) | 43.6 (11.6) | 41.0 (12.8) | 44.2 (15.3) | 46.2 (15.0) | 43.4 (15.9) | 44.7 (12.0) | 42.7 (11.2) |
| Hydrogenated oils | 0.6 (0.9) | 0.6 (0.9)¥ | 1.2 (2.9)¥ | 0.5 (0.6) | 0.6 (1.2) | 0.7 (0.8) | 0.5 (0.9) | 1.6 (4.1) | 0.8 (1.1) |
| Purchased deep fried foods | 47.8 (35.0)* | 35.8 (30.4)*¥ | 46.6 (31.6)¥ | 36.7 (29.2)‡ | 57.6 (37.0)‡ | 37.5 (34.8) | 34.2 (26.1) | 44.8 (34.0) | 48.0 (29.9) |

Footnote: Food groups are classified at the ingredient level, except for purchased deep fried foods which are assigned at the level of mixed dishes (ingredients in this food group are therefore "double-counted" as belonging to both purchased deep fried foods as well as to the underlying food groups that are fried). Food group definitions follow Table 4 in Bromage 2021 et al. [29], except for "Other fruits" (which was expanded to include coconuts in this study) and "Hydrogenated oils" (which are newly added in this study). p for difference <0.05 (t-test): * cases vs. HH controls, † cases vs. community controls, ¥ HH vs. community controls, ‡ male vs. female cases, § male vs. female HH controls, # male vs. female community controls. Abbreviations: HH, household; SSBs, sugar-sweetened beverages.

**Global Public Health**

**Table 3. Mean (SD) diet metric scores by population subgroup.**

| Metric | Cases, N=90 | HH controls, N=180 | Community controls, N=82 | Male cases, N=42 | Female cases, N=48 | Male HH controls, N=84 | Female HH controls, N=96 | Male community controls, N=37 | Female community controls, N=45 |
|---|---|---|---|---|---|---|---|---|---|
| GDQS | 22.0 (4.7)*† | 23.7 (4.1)* | 24.6 (4.1)† | 23.7 (4.5)‡ | 20.6 (4.4)‡ | 24.4 (3.9) | 23.2 (4.2) | 25.7 (4.1)# | 23.6 (4.0)# |
| GDQS, energy-adjusted | 21.0 (7.2)*† | 23.9 (6.4)* | 25.2 (6.9)† | 23.3 (7.0)‡ | 19.1 (6.9)‡ | 24.7 (6.3) | 23.3 (6.5) | 26.6 (6.8) | 24.0 (6.9) |
| GDQS+ | 11.0 (5.3)*† | 12.7 (4.7)*¥ | 14.0 (4.6)†¥ | 12.8 (5.0)‡ | 9.4 (5.0)‡ | 13.3 (4.7) | 12.1 (4.8) | 15.3 (4.8)# | 13.0 (4.2)# |
| GDQS+, energy-adjusted | 10.1 (7.5)*† | 12.7 (6.9)*¥ | 14.9 (6.9)†¥ | 12.4 (7.2)‡ | 8.1 (7.4)‡ | 13.3 (6.8) | 12.2 (6.9) | 16.4 (7.0) | 13.7 (6.7) |
| GDQS- | 11.0 (1.4)† | 11.1 (1.6)¥ | 10.5 (1.5)†¥ | 10.9 (1.4) | 11.1 (1.4) | 11.1 (1.8) | 11.1 (1.5) | 10.4 (1.9) | 10.6 (1.1) |
| GDQS-, energy-adjusted | 10.9 (2.4) | 11.2 (2.8)¥ | 10.2 (2.7)¥ | 10.9 (2.5) | 10.9 (2.3) | 11.4 (3.1) | 11.1 (2.6) | 10.2 (3.4) | 10.3 (2.0) |

Footnote: GDQS and GDQS sub-metrics computed following guidance from Bromage et al. 2021 [29]. Energy-adjusted metrics are computed using the residual method [45] (the mean metric score within each subgroup is added to the residuals for interpretability). p for difference <0.05 (t-test): * cases vs. HH controls, † cases vs. community controls, ¥ HH vs. community controls, ‡ male vs. female cases, § male vs. female HH controls, # male vs. female community controls. Abbreviations: HH, household; GDQS, Global Diet Quality Score; GDQS+, GDQS positive sub-metric; GDQS-, GDQS negative sub-metric.

analysis of intake densities (consumption per 2000 kcal), women had higher intake of purchased deep fried foods and white roots and tubers (albeit lower intake of juice and sugar-sweetened beverages) within cases; lower intake of poultry and higher intake of sweets and caloric sweeteners within both household and community controls; and lower intake of eggs within household controls (Table 2).

Diet quality. Cases had significantly (p<0.05) lower mean GDQS and GDQS+ (22.0 and 11.0, respectively) than both household (23.7 and 12.7) and community (24.6 and 14.0) controls (Table 3). Mean GDQS+ was also significantly lower in household than community controls (12.7 vs. 14.0, respectively) (Table 3). Community controls had significantly lower mean GDQS- (indicating higher consumption of unhealthy foods) than both cases (10.5 vs. 11.0, respectively) and household controls (10.5 vs. 11.1), although the former difference was not robust to energy adjustment (Table 3). Men had significantly higher GDQS and GDQS+ than women within both cases and community controls; sex differences in mean GDQS+ were not robust to energy adjustment (Table 3).

Individual and household food consumption-related habits. Among the total study population, 14.5% of the study population consumed meals purchased outside of the home more than once a week, 54.0% observed fasting (of which 79.5% fasted during a specified period of the year such as Ramadan or Paryushan), and 16.8% reported drinking alcohol (of which 49.2% drank at least once a week) (S1 Table). Among cases, 77.8% experienced some loss of appetite over the past month, of which 50.6% had experienced this every day (S1 Table).

## Nutrient intake and adequacy

Following examination of food consumption patterns, we examined nutrient intake (Table 4) and adequacy across cases and controls (Table 5).

Nutrient intake. Mean energy intake was 1957.0 kcal in cases, 2185.3 kcal in household controls, and 2336.4 kcal in community controls (p<0.05 between cases and community controls) (Table 4). Among cases, the breakdown of caloric energy from different macronutrients was 11.8% protein, 29.8% fat (of which 25.0% was saturated, 22.6% monounsaturated, 41.2% polyunsaturated, and 0.3% *trans*), 57.9% carbohydrate, and 1.7% alcohol (Table 4). These proportions did not differ significantly between study arms except for a small difference in the percentage of energy from monounsaturated fats between household (23.1%) vs. community (22.6%) controls (Table 4). Men consumed a significantly higher proportion of energy than women from protein within cases, monounsaturated fat within household controls, and alcohol within both cases and community controls; and a significantly lower proportion of energy from

**Table 4. Mean (SD) daily nutrient intake by population subgroup.**

| Nutrient | Cases, N=90 | HH controls, N=180 | Community controls, N=82 | Male cases, N=42 | Female cases, N=48 | Male HH controls, N=84 | Female HH controls, N=96 | Male community controls, N=37 | Female community controls, N=45 |
|---|---|---|---|---|---|---|---|---|---|
| Energy (kcal) | 1957 (1248)† | 2185 (1221) | 2336 (1233)† | 2377 (1308)‡ | 1589 (1077)‡ | 2397 (1295)‡ | 2001 (1126)‡ | 2678 (1424)‡ | 2056 (980)‡ |
| Protein (g) | 59.2 (45.3) | 66.3 (44.1) | 68.2 (38.9) | 74.1 (49.9)‡ | 46.1 (36.7)‡ | 73.4 (44.5)‡ | 60.1 (43.0)‡ | 79.0 (46.7)‡ | 59.4 (28.7)‡ |
| g/kg | 1.3 (1.1) | 1.2 (0.8) | 1.2 (0.6) | 1.5 (1.1) | 1.2 (1.0) | 1.3 (0.9) | 1.1 (0.8) | 1.2 (0.7) | 1.1 (0.5) |
| % TEI | 11.8 (2.4) | 11.9 (2.1) | 11.7 (1.9) | 12.3 (2.5)‡ | 11.3 (2.1)‡ | 12.2 (2.2) | 11.7 (2.0) | 11.8 (2.0) | 11.6 (1.8) |
| Fat (g) | 67.4 (52.6) | 74.0 (53.5) | 79.0 (43.8) | 81.2 (58.1)‡ | 55.3 (44.5)‡ | 81.9 (57.1) | 67.1 (49.5) | 91.3 (49.4)‡ | 69.0 (36.1)‡ |
| % TEI | 29.8 (7.2) | 29.3 (7.4) | 30.5 (6.8) | 29.7 (7.7) | 30.0 (6.7) | 30.1 (8.1) | 28.6 (6.8) | 31.2 (7.0) | 29.9 (6.7) |
| % saturated | 25.0 (5.3) | 25.4 (5.4) | 25.1 (4.9) | 25.2 (5.0) | 24.9 (5.6) | 25.5 (5.1) | 25.3 (5.7) | 25.6 (5.7) | 24.6 (4.1) |
| % monounsaturated | 22.6 (2.3) | 23.1 (2.0)¥ | 22.6 (1.8)¥ | 22.7 (2.6) | 22.5 (1.9) | 23.5 (1.9)‡ | 22.8 (2.0)‡ | 22.8 (1.9) | 22.4 (1.7) |
| % polyunsaturated | 41.2 (7.1) | 42.0 (7.2) | 42.5 (5.3) | 40.8 (7.6) | 41.6 (6.7) | 42.1 (6.7) | 41.9 (7.8) | 42.5 (5.7) | 42.4 (4.9) |
| % trans | 0.3 (0.2) | 0.4 (0.3) | 0.4 (0.7) | 0.2 (0.2)‡ | 0.3 (0.2)‡ | 0.4 (0.4) | 0.3 (0.3) | 0.5 (1.0) | 0.3 (0.4) |
| Cholesterol (mg) | 325.3 (416.6) | 337.3 (339.0) | 344.4 (281.2) | 447.5 (513.2)‡ | 218.4 (271.6)‡ | 416.4 (382.5)‡ | 268.1 (279.9)‡ | 429.5 (307.6)‡ | 274.4 (238.9)‡ |
| Carbohydrate (g) | 275.5 (166.5)† | 313.5 (163.2) | 334.0 (184.3)† | 329.8 (174.8)‡ | 227.9 (144.5)‡ | 340.1 (180.3)‡ | 290.3 (143.6)‡ | 374.4 (214.2) | 300.7 (149.9) |
| % TEI | 57.9 (9.0) | 58.8 (8.8) | 57.5 (8.0) | 56.6 (9.9) | 59.0 (8.2) | 57.5 (9.5) | 59.9 (8.0) | 55.9 (8.2) | 58.8 (7.6) |
| Alcohol (g) | 4.2 (13.8)* | 1.2 (5.4)*¥ | 4.7 (15.3)¥ | 8.8 (19.3)‡ | 0.1 (0.1)‡ | 2.4 (7.7)‡ | 0.2 (0.4)‡ | 10.3 (21.7)‡ | 0.2 (0.2)‡ |
| % TEI | 1.7 (6.7) | 0.6 (4.0) | 1.0 (3.3) | 3.6 (9.4)‡ | 0.1 (0.0)‡ | 1.2 (5.8) | 0.1 (0.2) | 2.2 (4.6)‡ | 0.1 (0.1)‡ |
| Fiber (g) | 7.6 (5.3)† | 8.9 (5.5) | 9.9 (5.2)† | 8.5 (5.1) | 6.8 (5.5) | 9.1 (5.2) | 8.7 (5.8) | 10.9 (5.5) | 9.1 (4.7) |
| Calcium (mg) | 687.5 (604.9) | 759.3 (633.9) | 725.7 (429.0) | 872.7 (714.5)‡ | 525.4 (436.0)‡ | 817.3 (671.1) | 708.5 (598.4) | 834.0 (489.5)‡ | 636.6 (353.3)‡ |
| Copper (mg) | 1.7 (1.2)*† | 2.0 (1.3)* | 2.2 (1.1)† | 2.0 (1.2)‡ | 1.4 (1.1)‡ | 2.2 (1.2) | 1.9 (1.4) | 2.5 (1.2)‡ | 1.9 (0.9)‡ |
| Iodine (µg) | 99.3 (79.1) | 109.5 (73.9) | 110.9 (68.0) | 126.5 (92.3)‡ | 75.6 (56.4)‡ | 125.7 (81.3)‡ | 95.3 (63.9)‡ | 131.9 (77.6)‡ | 93.6 (53.8)‡ |
| Iron (mg) | 14.1 (11.8) | 16.3 (12.2) | 16.8 (9.0) | 17.8 (13.4)‡ | 10.9 (9.1)‡ | 17.9 (12.4) | 14.9 (11.9) | 19.4 (10.1)‡ | 14.6 (7.4)‡ |
| Magnesium (mg) | 394.5 (251.4)*† | 489.2 (323.3)* | 506.6 (261.7)† | 482.2 (271.3)‡ | 317.7 (206.3)‡ | 523.3 (338.5) | 459.3 (308.1) | 561.5 (272.0) | 461.4 (246.7) |
| Manganese (mg) | 4.4 (2.5)*† | 5.4 (2.9)* | 5.5 (2.9)† | 5.3 (2.7)‡ | 3.6 (1.9)‡ | 5.7 (2.7) | 5.2 (3.0) | 6.2 (3.5) | 4.9 (2.3) |
| Phosphorous (mg) | 1145.2 (830.5) | 1331.8 (813.8) | 1372.5 (725.8) | 1441.2 (927.1)‡ | 886.2 (639.9)‡ | 1481.2 (843.3)‡ | 1201.0 (767.7)‡ | 1595.5 (845.7)‡ | 1189.2 (555.3)‡ |
| Potassium (mg) | 1878.7 (1224.5)† | 2188.3 (1258.5) | 2263.4 (1178.4)† | 2207.7 (1267.1)‡ | 1590.9 (1121.4)‡ | 2328.0 (1289.0) | 2066.0 (1224.9) | 2618.4 (1381.6)‡ | 1971.5 (895.3)‡ |
| Selenium (µg) | 91.8 (85.6) | 105.0 (85.8) | 109.6 (71.0) | 124.7 (103.0)‡ | 63.1 (53.1)‡ | 119.8 (89.4)‡ | 92.1 (80.8)‡ | 124.3 (79.5) | 97.5 (61.6) |
| Sodium (mg) | 4208.0 (3552.6)† | 4820.0 (3572.8) | 5334.1 (3762.5)† | 5056.0 (3809.6)‡ | 3465.9 (3167.6)‡ | 5249.2 (3502.5) | 4444.4 (3609.6) | 6309.8 (4730.1)‡ | 4531.9 (2509.2)‡ |
| Zinc (mg) | 7.4 (5.0)† | 8.6 (4.8) | 8.9 (4.8)† | 9.1 (5.3)‡ | 5.9 (4.1)‡ | 9.5 (4.9)‡ | 7.8 (4.5)‡ | 10.2 (5.5)‡ | 7.9 (3.8)‡ |
| Vitamin A (µg RAE) | 680.1 (566.9)† | 835.1 (651.9) | 943.5 (967.5)† | 841.6 (595.8)‡ | 538.9 (505.3)‡ | 911.4 (688.4) | 768.4 (614.1) | 1113.8 (1304.8) | 803.5 (534.5) |
| Thiamine (mg) | 0.9 (0.6)*† | 1.1 (0.7)* | 1.2 (0.6)† | 1.1 (0.6)‡ | 0.7 (0.5)‡ | 1.2 (0.7)‡ | 1.0 (0.6)‡ | 1.4 (0.7)‡ | 1.1 (0.5)‡ |
| Riboflavin (mg) | 1.1 (0.8) | 1.3 (0.8) | 1.3 (0.7) | 1.4 (0.9)‡ | 0.9 (0.6)‡ | 1.4 (0.9)‡ | 1.2 (0.7)‡ | 1.5 (0.8)‡ | 1.1 (0.6)‡ |
| Niacin (mg) | 12.5 (9.9) | 14.3 (10.2) | 14.9 (9.5) | 15.6 (10.5)‡ | 9.8 (8.4)‡ | 15.5 (10.1) | 13.1 (10.3) | 17.2 (11.9)‡ | 13.0 (6.4)‡ |
| Pantothenic acid (mg) | 5.1 (3.2)† | 5.9 (3.0) | 6.2 (3.5)† | 6.2 (3.3)‡ | 4.2 (2.9)‡ | 6.6 (3.2)‡ | 5.3 (2.7)‡ | 7.1 (4.0)‡ | 5.4 (2.8)‡ |

*(Continued)*

**Table 4.** (Continued)

| Nutrient | Cases, N = 90 | HH controls, N = 180 | Community controls, N = 82 | Male cases, N = 42 | Female cases, N = 48 | Male HH controls, N = 84 | Female HH controls, N = 96 | Male community controls, N = 37 | Female community controls, N = 45 |
|---|---|---|---|---|---|---|---|---|---|
| Vitamin B6 (mg) | 1.4 (1.1)† | 1.7 (0.9) | 1.8 (1.3)† | 1.8 (1.2)‡ | 1.2 (0.9)‡ | 1.9 (1.0)‡ | 1.5 (0.9)‡ | 2.2 (1.8)‡ | 1.5 (0.7)‡ |
| Folate (µg) | 245.6 (167.2)† | 282.9 (161.2) | 305.8 (156.8)† | 288.4 (175.0)‡ | 208.2 (152.3)‡ | 312.6 (175.6)‡ | 256.9 (143.3)‡ | 353.2 (177.5)‡ | 266.8 (126.8)‡ |
| Vitamin B12 (µg) | 2.4 (3.2) | 2.5 (3.2) | 2.3 (2.8) | 3.2 (3.0)‡ | 1.7 (3.1)‡ | 2.7 (3.2) | 2.2 (3.2) | 2.3 (1.6) | 2.2 (3.6) |
| Vitamin C (mg) | 82.5 (76.0)† | 97.8 (69.1) | 115.5 (97.4)† | 90.3 (70.6) | 75.6 (80.4) | 99.6 (72.0) | 96.3 (66.7) | 129.5 (102.9) | 104.0 (92.2) |
| Vitamin E (mg ATE) | 22.8 (18.8) | 25.4 (19.4) | 27.2 (14.6) | 27.1 (20.7)‡ | 19.0 (16.2)‡ | 27.9 (20.2) | 23.2 (18.6) | 31.1 (16.2)‡ | 24.0 (12.4)‡ |
| Vitamin D (µg) | 0.8 (1.1) | 0.8 (0.9) | 0.9 (0.9) | 1.1 (1.3)‡ | 0.5 (0.6)‡ | 1.1 (1.1)‡ | 0.6 (0.7)‡ | 1.1 (1.0) | 0.7 (0.8) |

Footnote: p for difference <0.05 (t-test): * cases vs. HH controls, † cases vs. community controls, ¥ HH vs. community controls, ‡ male vs. female cases, § male vs. female HH controls, # male vs. female community controls. Abbreviations: HH, household; TEI, total energy intake; RAE, retinol activity equivalents; ATE, alpha-tocopherol equivalents.

*trans* fat within cases (Table 4). Among the total study population, only 4.5% used any vitamin or mineral supplements (S1 Table).

Nutrient adequacy. Among cases, the prevalence of nutrient adequacy exceeded 50% for only six nutrients: selenium (73%), protein (72%), sodium (69%), vitamin E (62%), and vitamin A and magnesium (each 57%) (Table 5). More than one-third but less than half of cases had adequate intake of vitamin C (46%), phosphorous (44%), manganese and folate (43%), niacin and pantothenic acid (41%), iodine (37%), vitamin B6 (36%), and copper (34%). For seven nutrients, adequacy prevalence did not exceed one-third in cases: vitamin B12 (33%), thiamine (31%), calcium (27%), iron (24.2%) riboflavin (20%), zinc (14%), potassium (11%), and fiber and vitamin D (0%) (Table 5). For all nutrients, the prevalence of adequacy in cases was either significantly lower than household or community controls (p<0.05) or did not differ, and was either significantly higher in community than household controls or did not differ (Table 5). Adequacy prevalence in men significantly exceeded that in women for all nutrients within cases; copper, iodine, iron, phosphorous, and vitamin B12 within household controls; and iron, phosphorous, and vitamin B6 within community controls (Table 5).

## Nutrition status

We also examined nutritional status among the study population (Table 6). Compared to both household and community controls, cases had significantly (p<0.05) lower mean BMI (17.3 vs. 23.4 and 23.9 kg/m² in cases vs. household and community controls, respectively), MUAC (23.4 vs. 28.3 and 28.5 cm), waist circumference (71.6 vs. 84.0 and 82.1 cm), hemoglobin (13.4 vs. 14.0 and 14.4 g/dl), and serum 25(OH)D (14.6 vs. 19.0 and 17.0 ng/ml); significantly higher prevalence of underweight BMI (66% vs. 23% and 15%), low MUAC (69% vs. 25% and 21%), anemia (22% vs. 9% and 10%), and diabetes (18% vs. 4% and 5%, respectively); and significantly lower prevalence of abdominal obesity (3% vs. 27 and 18%, respectively) (Table 6). The prevalence of heart disease and hypertension in cases (2% and 4%, respectively) did not differ significantly from either household (2% and 5%) or community (1% and 5%) controls (Table 6). No significant differences in any anthropometric, biochemical, or clinical measures of nutrition status were observed between household and community controls (Table 6).

The prevalence of vitamin D deficiency in women significantly exceeded that in men within all three study arms (Table 6). Men had a significantly higher prevalence of underweight and a lower prevalence of abdominal obesity, anemia, and

**Table 5. Prevalence of nutrient adequacy by population subgroup, n (%).**

| Nutrient | Cases, N = 90 | HH controls, N = 180 | Community controls, N = 82 | Male cases, N = 42 | Female cases, N = 48 | Male HH controls, N = 84 | Female HH controls, N = 96 | Male community controls, N = 37 | Female community controls, N = 45 |
|---|---|---|---|---|---|---|---|---|---|
| Protein (g/kg) | 65 (72.2)† | 130 (72.2)¥ | 70 (85.4)†¥ | 35 (83.3)‡ | 30 (62.5)‡ | 66 (78.6) | 64 (66.7) | 34 (91.9) | 36 (80.0) |
| Fiber | 0 (0.0) | 2 (1.1) | 0 (0.0) | 0 (0.0) | 0 (0.0) | 0 (0.0) | 2 (2.1) | 0 (0.0) | 0 (0.0) |
| Calcium | 24 (26.7) | 51 (28.3) | 22 (26.8) | 17 (40.5)‡ | 7 (14.6)‡ | 26 (31.0) | 25 (26.0) | 12 (32.4) | 10 (22.2) |
| Copper | 31 (34.4)† | 70 (38.9) | 41 (50.0)† | 19 (45.2)‡ | 12 (25.0)‡ | 41 (48.8)‡ | 29 (30.2)‡ | 23 (62.2) | 18 (40.0) |
| Iodine | 33 (36.7) | 82 (45.6) | 41 (50.0) | 21 (50.0)‡ | 12 (25.0)‡ | 50 (59.5)‡ | 32 (33.3)‡ | 23 (62.2) | 18 (40.0) |
| Iron | n.a. (24.2) | n.a. (29.3) | n.a. (30.3) | n.a. (26.8) | n.a. (21.9) | n.a. (25.2) | n.a. (32.9) | n.a. (30.5) | n.a. (30.2) |
| Magnesium | 51 (56.7)*† | 135 (75.0)* | 69 (84.1)† | 30 (71.4)‡ | 21 (43.8)‡ | 62 (73.8) | 73 (76.0) | 31 (83.8) | 38 (84.4) |
| Manganese | 39 (43.3)*† | 121 (67.2)* | 54 (65.9)† | 26 (61.9)‡ | 13 (27.1)‡ | 60 (71.4) | 61 (63.5) | 26 (70.3) | 28 (62.2) |
| Phosphorous | 40 (44.4)*† | 107 (59.4)* | 54 (65.9)† | 27 (64.3)‡ | 13 (27.1)‡ | 59 (70.2)‡ | 48 (50.0)‡ | 30 (81.1)‡ | 24 (53.3)‡ |
| Potassium | 10 (11.1) | 22 (12.2) | 8 (9.8) | 6 (14.3) | 4 (8.3) | 13 (15.5) | 9 (9.4) | 5 (13.5) | 3 (6.7) |
| Selenium | 66 (73.3)*† | 159 (88.3)* | 75 (91.5)† | 38 (90.5)‡ | 28 (58.3)‡ | 76 (90.5) | 83 (86.5) | 36 (97.3) | 39 (86.7) |
| Sodium | 62 (68.9)*† | 154 (85.6)*¥ | 78 (95.1)†¥ | 34 (81.0)‡ | 28 (58.3)‡ | 76 (90.5) | 78 (81.3) | 36 (97.3) | 42 (93.3) |
| Zinc | 13 (14.4) | 30 (16.7) | 11 (13.4) | 7 (16.7) | 6 (12.5) | 13 (15.5) | 17 (17.7) | 4 (10.8) | 7 (15.6) |
| Vitamin A | 51 (56.7)*† | 130 (72.2)*¥ | 70 (85.4)†¥ | 30 (71.4)‡ | 21 (43.8)‡ | 64 (76.2) | 66 (68.8) | 33 (89.2) | 37 (82.2) |
| Thiamine | 28 (31.1)† | 63 (35.0)¥ | 40 (48.8)†¥ | 18 (42.9)‡ | 10 (20.8)‡ | 30 (35.7) | 33 (34.4) | 21 (56.8) | 19 (42.2) |
| Riboflavin | 18 (20.0) | 35 (19.4) | 19 (23.2) | 12 (28.6) | 6 (12.5) | 19 (22.6) | 16 (16.7) | 12 (32.4) | 7 (15.6) |
| Niacin | 37 (41.1)† | 96 (53.3) | 53 (64.6)† | 23 (54.8)‡ | 14 (29.2)‡ | 41 (48.8) | 55 (57.3) | 23 (62.2) | 30 (66.7) |
| Pantothenic acid | 37 (41.1)* | 99.0 (55.0)* | 44 (53.7) | 24 (57.1)‡ | 13 (27.1)‡ | 51 (60.7) | 48 (50.0) | 23 (62.2) | 21 (46.7) |
| Vitamin B6 | 32 (35.6) | 75 (41.7) | 37 (45.1) | 20 (47.6)‡ | 12 (25.0)‡ | 39 (46.4) | 36 (37.5) | 22 (59.5)‡ | 15 (33.3)‡ |
| Folate | 39 (43.3)*† | 109 (60.6)*¥ | 61 (74.4)†¥ | 21 (50.0) | 18 (37.5) | 49 (58.3) | 60 (62.5) | 28 (75.7) | 33 (73.3) |
| Vitamin B12 | 30 (33.3) | 63 (35.0) | 27 (32.9) | 21 (50.0)‡ | 9 (18.8)‡ | 36 (42.9)‡ | 27 (28.1)‡ | 14 (37.8) | 13 (28.9) |
| Vitamin C | 41 (45.6)*† | 122 (67.8)* | 65 (79.3)† | 24 (57.1) | 17 (35.4) | 54 (64.3) | 68 (70.8) | 31 (83.8) | 34 (75.6) |
| Vitamin E | 56 (62.2)*† | 144 (80.0)*¥ | 76 (92.7)†¥ | 32 (76.2)‡ | 24 (50.0)‡ | 71 (84.5) | 73 (76.0) | 35 (94.6) | 41 (91.1) |
| Vitamin D | 0 (0.0) | 0 (0.0) | 0 (0.0) | 0 (0.0) | 0 (0.0) | 0 (0.0) | 0 (0.0) | 0 (0.0) | 0 (0.0) |

Footnote: For nutrients except iron, prevalence of nutrient adequacy is computed as the percentage of participants with intake meeting Estimated Average Requirements (EARs) [33] (or Recommended Intake Levels, when EARs are unavailable) from the Nutrient Requirements for Indians (NRIs) [34] (or Adequate Intake Levels from the US Dietary Reference Intakes [33], when no level is specified in the NRIs). Probability of iron adequacy is computed using the full probability approach [33]. Color coding: darkest red, lowest observed prevalence (0%); brightest yellow: 50th percentile of observed prevalence (48.2%), darkest green: highest observed prevalence (97.3%). p for difference <0.05 (Fisher's exact test). * cases vs. HH controls, † cases vs. community controls, ¥ HH vs. community controls, ‡ male vs. female cases, § male vs. female HH controls, # male vs. female community controls. Abbreviations: HH, household; n.a., not applicable.

hypertension than women within household controls; and a higher prevalence of type 2 diabetes within community controls (Table 6).

## Adjusted trends across age and asset categories

Finally, we examined trends in food and nutrition consumption patterns across age and asset categories (Figs 1, 2, S2 Table).

Adjusting for age category, asset quartile, sex, and study arm, significant (p < 0.05) linear trends were observed in estimated marginal mean intake of all selected food groups and nutrients across age categories (except for total animal-source foods, oils, calcium, iron, selenium, and vitamin B12) and across asset quartiles (except for total grains and flour products, alcohol, and vitamin D) (Figs 1A-1C, 2A-2C, S2 Table). Most trends across increasing age categories were

**Table 6. Anthropometric, biochemical, and clinical measures of nutrition status by population subgroup.**

| Outcome | Cases, | HH controls, | Community | Male cases, | Female | Male HH | Female HH | Male community | Female |
|---|---|---|---|---|---|---|---|---|---|
| | N=90 | N=180 | controls, N=82 | N=42 | cases, N=48 | controls, N=84 | controls, N=96 | controls, N=37 | community controls, N=45 |
| BMI (kg/m²), mean (SD) | 17.7 (3.3)*† | 23.4 (5.5)* | 23.9 (5.1)† | 17.6 (3.5) | 17.7 (3.3) | 22.0 (4.4)§ | 24.7 (6.1)§ | 24.6 (5.1) | 23.3 (5.0) |
| <18.5, n (%) | 59 (65.6)*† | 41 (22.8)* | 12 (14.6)† | 28 (66.7) | 31 (64.6) | 22 (26.2)§ | 19 (19.8)§ | 4 (10.8) | 8 (17.8) |
| 18.5-22.9, n (%) | 23 (25.6) | 52 (28.9) | 28 (34.1) | 11 (26.2) | 12 (25.0) | 29 (34.5) | 23 (24.0) | 11 (29.7) | 17 (37.8) |
| 23-24.9, n (%) | 5 (5.6) | 23 (12.8) | 11 (13.4) | 1 (2.4) | 4 (8.3) | 12 (14.3) | 11 (11.5) | 8 (21.6) | 3 (6.7) |
| ≥25, n (%) | 3 (3.3) | 64 (35.6) | 31 (37.8) | 2 (4.8) | 1 (2.1) | 21 (25.0) | 43 (44.8) | 14 (37.8) | 17 (37.8) |
| MUAC (cm), mean (SD) | 23.4 (3.6)*† | 28.3 (4.3)* | 28.5 (4.6)† | 24.7 (3.4)‡ | 22.3 (3.5)‡ | 28.3 (3.4) | 28.4 (5.0) | 30.6 (4.1)# | 26.8 (4.4)# |
| Low MUAC, n (%) | 62 (68.9)*† | 45 (25.0)* | 17 (20.7)† | 25 (59.5) | 37 (77.1) | 20 (23.8) | 25 (26.0) | 4 (10.8) | 13 (28.9) |
| Waist circumference (cm), mean (SD) | 71.6 (10.9)*† | 84 (13.2)* | 82.1 (13.6)† | 74.2 (10.4)‡ | 69.3 (10.9)‡ | 84.3 (11.9) | 82.9 (14.2) | 86.6 (13.4)# | 78.4 (12.8)# |
| Abdominal obesity, n (%) | 3 (3.3)*† | 49.0 (27.2)* | 15 (18.3)† | 0 (0.0) | 3 (6.3) | 6 (7.1)§ | 43 (44.8)§ | 5 (13.5) | 10 (22.2) |
| Hemoglobin (g/dl), mean (SD) | 13.4 (1.6)*† | 14 (1.8)* | 14.4 (1.7)† | 14.4 (1.4)‡ | 12.6 (1.3)‡ | 15.5 (1.2)§ | 13.3 (1.5)§ | 15.6 (1.2)# | 13.4 (1.3)# |
| Anemia, n (%) | 20 (22.2)*† | 17.0 (9.4)* | 8 (9.8)† | 8 (19.0) | 12 (25.0) | 3 (3.6)§ | 14 (14.6)§ | 1 (2.7) | 7 (15.6) |
| Glucose (mg/dl), mean (SD) | 111.4 (52.8) | 106.6 (53.9) | 105.0 (43.3) | 120.9 (67.9) | 103.0 (33.2) | 110.7 (69.3) | 103.0 (35.4) | 116.4 (61.0)# | 95.7 (14.3)# |
| Serum 25(OH)D (ng/ml), mean (SD) | 14.6 (12.8)* | 19 (13.8)* | 17.0 (10.5)† | 18.2 (13.3)‡ | 11.6 (11.7)‡ | 20.0 (12.3) | 18.2 (15.0) | 19.0 (10.6) | 15.3 (10.1) |
| 0 -<10, n (%) | 40 (44.4) | 57 (31.7) | 23 (28.0) | 14 (33.3)‡ | 26 (54.2)‡ | 18 (21.4)§ | 39 (40.6)§ | 3 (8.1)# | 20 (44.4)# |
| 10 -<20, n (%) | 32 (35.6) | 59 (32.8) | 34 (41.5) | 14 (33.3) | 18 (37.5) | 35 (41.7) | 24 (25.0) | 23 (62.2) | 11 (24.4) |
| 20 -<30, n (%) | 9 (10.0) | 32 (17.8) | 17 (20.7) | 7 (16.7) | 2 (4.2) | 15 (17.9) | 17 (17.7) | 7 (18.9) | 10 (22.2) |
| ≥30, n (%) | 9 (10.0) | 32 (17.8) | 8 (9.8) | 7 (16.7) | 2 (4.2) | 16 (19.0) | 16 (16.7) | 4 (10.8) | 4 (8.9) |
| Heart disease, n (%) | 2 (2.2) | 3 (1.7) | 1 (1.2) | 1 (2.4) | 1 (2.1) | 1 (1.2) | 2 (2.1) | 1 (2.7) | 0 (0.0) |
| Hypertension, n (%) | 4 (4.4) | 9 (5.0) | 4 (4.9) | 2 (4.8) | 2 (4.2) | 1 (1.2)§ | 8 (8.3)§ | 2 (5.4) | 2 (4.4) |
| Diabetes, n (%) | 16 (17.8)*† | 8 (4.4)* | 4 (4.9)† | 10 (23.8) | 6 (12.5) | 5 (6.0) | 3 (0.1) | 4 (10.8)# | 0 (0.0)# |

Footnote: Low MUAC defined as <25.5 cm in men and <24.5 cm in women [39]. Abdominal obesity defined as waist circumference >102 cm in men and >88 cm in women [38]. Anemia defined as <12 g/dl hemoglobin in women and <13 g/dl in men [40]. Glucose measurements are taken at random times during the day. Data on heart disease, hypertension, and diabetes were obtained from medical records and self-reports for cases and controls, respectively. p for difference <0.05 (t-test for continuous variables, Fisher's exact test for categorical variables): * cases vs. HH controls, † cases vs. community controls, ¥ HH vs. community controls, ‡ male vs. female cases, § male vs. female HH controls, # male vs. female community controls. Abbreviations: HH, household; BMI, body mass index; MUAC, mid-upper arm circumference; 25(OH)D, 25-hydroxyvitamin D.

secular and negative, while most trends across increasing asset quartiles were secular and positive (for all nonsecular trends, the estimated marginal mean in the youngest age category exceeded that in the oldest, and the estimated marginal mean in the highest asset quartile exceeded that in the lowest) (Figs 1A-1C, 2A-2C, S2 Table). In the majority of cases, there was no clear evidence of interaction between household case/control status and age, asset quartile, or sex (S2 Table).

Controlling for age category, sex, and study arm, positive secular trends in GDQS and GDQS +, and negative secular trends in GDQS- (indicating higher consumption of unhealthy foods) were observed across increasing quartiles of asset index (p for linear trend <0.05) (Figs 1D, 2D, S2 Table). Controlling for asset quartile, sex, and study arm, a significant non-secular negative trend in GDQS+ and a positive secular trend in GDQS- were also observed across increasing age categories, although only the latter was robust to adjustment for total energy intake (S2 Table). No significant interactions between study arm and either age, sex, or asset quartile were observed for dietary metric outcomes (S2 Table).

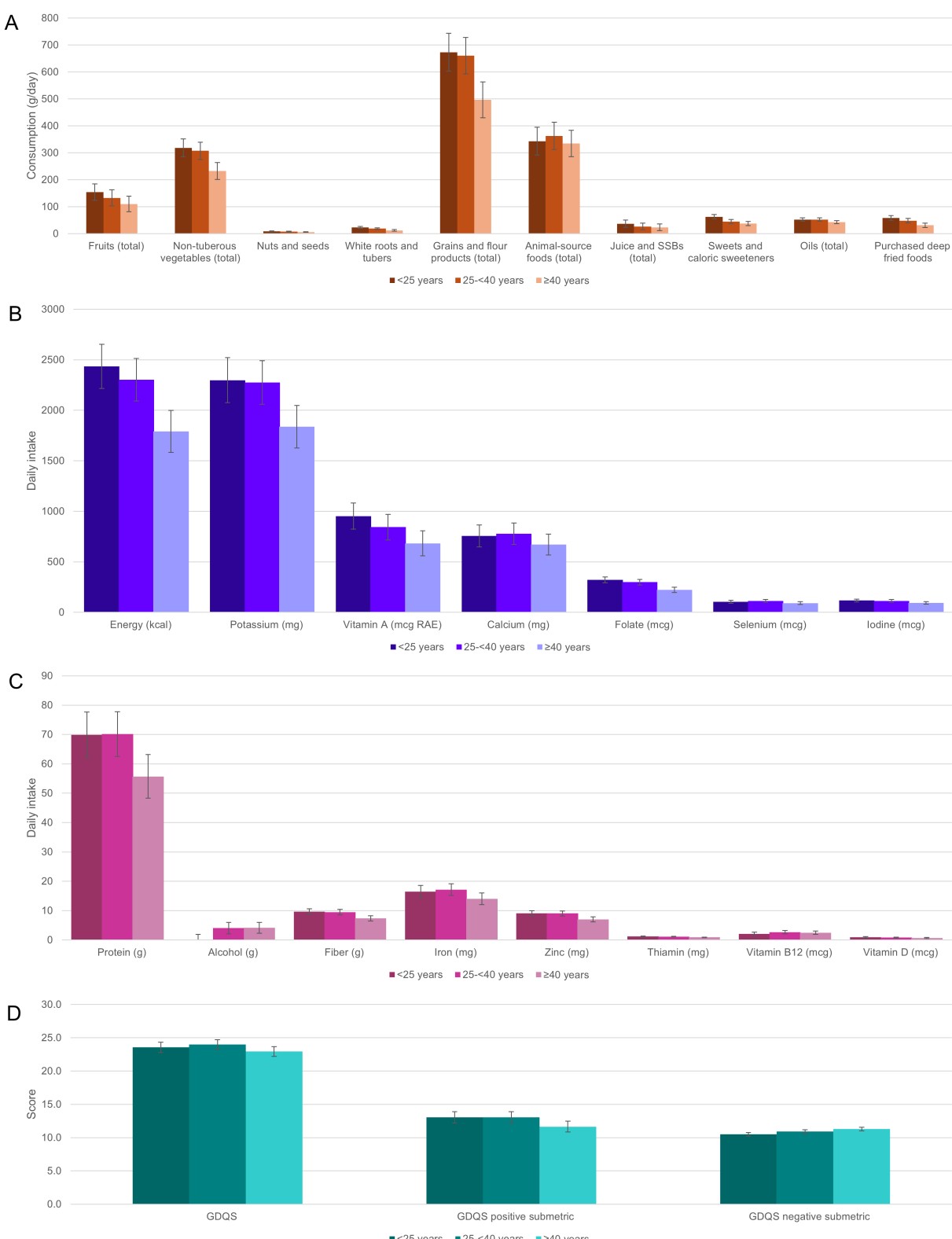

**Fig 1. Estimated marginal mean (95%CI) consumption of key food groups (A) and nutrients (B, C), and diet metric scores (D) by age category.**

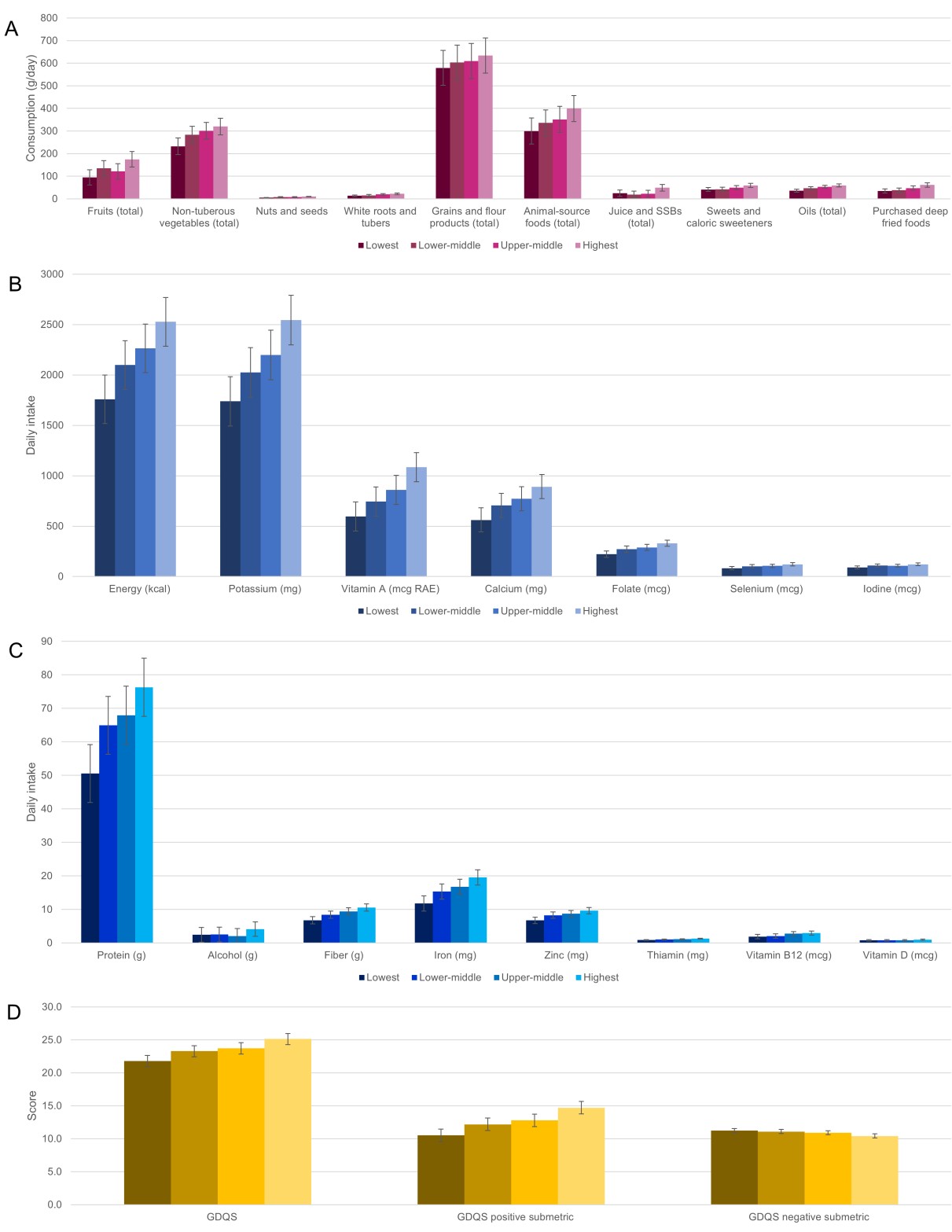

**Fig 2. Estimated marginal mean (95%CI) consumption of key food groups (A) and nutrients (B, C), and diet metric scores (D) by asset quartile.**

## Discussion

This analysis of slum residents in Mumbai found a high prevalence of multiple nutrient inadequacies and clinical malnutrition among adult pulmonary MDR-TB cases, household controls, and community controls, and notable disparities in diet and nutrition status between cases and controls. Compared to controls, cases had lower consumption of major healthy food groups including non-tuberous and deep orange vegetables, legumes, whole grains, nuts and seeds; lower GDQS and GDQS+ scores; higher prevalence of dietary nutrient inadequacies including protein, thiamine, folate, and vitamins A, C, and E; and higher burdens of underweight, anemia, and type 2 diabetes.

These findings align with prior surveys of drug-sensitive (DS) and MDR-TB in India. A 2011–2016 survey comparing diet and nutritional parameters among 747 healthy adult controls and pulmonary DS-TB and MDR-TB cases in New Delhi found lower BMI and intake of all macronutrients, P, Ca, and vitamin D in MDR-TB as compared to DS-TB cases and DS-TB cases as compared to controls [46]. Among 1,695 adult pulmonary TB cases in Chhattisgarh from 2004-2009, 89% were underweight and 52% stunted at diagnosis, and each one-unit higher BMI was associated with a 0.78 adjusted odds ratio of TB death [47]. Population attributable fraction analysis of NFHS-3 data has indicated undernutrition is a major risk factor for TB infection and mortality in India, where it is implicated in 53.9% and 55.8% of deaths in men and women of reproductive age, respectively [48]. Nutritional support has conversely been associated with substantially decreased TB mortality [49,50], including in a cohort of 2,800 Indian adults [49]. These findings suggest that nutritional interventions may mitigate disease severity, enhance immune recovery, and improve treatment outcomes in MDR-TB specifically, beyond their general benefits in TB care.

The constellation of dietary risk factors and manifestations of malnutrition that we observed among MDR-TB cases contribute to, perpetuate, and result from the disease. Imbalances in nutrient intake impair cell-mediated immunity, weaken mucosal barriers, alter gut microbiome, and increase oxidative stress [14,15]. Dietary imbalances also contribute to metabolic disfunction and low-grade inflammation, which contribute in turn to impaired immune and lung function [14,15,19,20]. These effects increase the risk of acquiring latent TB infection and progression to active disease. In active TB, poor diet and nutrition lead to poorer treatment outcomes by continuing to impair immune response, delaying healing and recovery, and reducing treatment efficacy [13–18]. TB itself exacerbates dietary deficiencies and malnutrition by reducing appetite and absorption, and increasing metabolic demands and catabolic activity [14–18]. This cycle is most pronounced in MDR-TB, in which immune response is especially impaired and treatment options are limited, less effective, have severe side effects, and are longer in duration, creating an environment inconducive to maintaining adequate nutrition and effectively fighting infection [10,11,13,17,51–53].

Although nutritional deficits and their impacts are magnified in those with TB, dietary and socio- economic risk factors are widespread in the source population [54–56], particularly among women and the poor. In the current study, women had poorer nutrient adequacy than men in all three study arms, and nutrient intake declined with asset index upon adjustment for age, sex, and study arm. A prior 2006–2012 survey among 6,426 slum-dwelling non-pregnant women of reproductive age (NP WRA) in Mumbai found 32.1% to be underweight and >50% inadequacy of dietary energy, Ca, Fe, Mg, Se, Zn, and vitamins A, B2, B3, B9, B12 [51]. Persisting nutritional inadequacies result from a predominance of nutrient-scarce grains, roots, and tubers in the diet (contributing 59.6% of total energy intake in the current study population, and 54.1% among urban adults nationally) [28] but are also the product of an exceedingly inequitable and insecure nutrition environment that is further implicated in the high burden of TB. For example, a 2015–2018 survey of 765 pulmonary TB cases ≥15 years in South India found a 34.1% prevalence of household food insecurity (HFI), of which 23.1% was severe [57], and analysis of 178 child TB cases and controls in Pune found a 11.55 adjusted odds ratio of confirmed/probable TB associated with HFI [58]. As in other countries, the poor in India are disproportionately affected by TB and nutrition insecurity [59–61], and in the current study, we found poorer overall diet quality and nutrient adequacy among those in the lowest wealth quartile.

Research on nutritional epidemiology of TB in low- and middle-income countries has focused heavily on the roles of dietary deficits and underweight, while less focus has been paid to the emerging importance of lifestyle risk factors for

(and manifestations of) diet-related non-communicable disease, of which type 2 diabetes is of particular concern in India [62,63]. Despite having higher rates of underweight, MDR-TB cases in the current study also had a higher prevalence of diabetes. Large burdens of both diabetes and undernutrition were also found among 173 adult pulmonary TB cases in Puducherry from 2017-2018, among whom a 48.8% prevalence of diabetes coexisted with 41.6% underweight, 63.0% anemia, and >70% dietary inadequacy of energy, protein, Fe, Ca, vitamins C and B12, and folate [64]. Diabetes is also a risk factor for TB infection: a 2013–2020 survey of 2,351 adults in southern India found no association between BMI and latent TB, but higher odds of TB in diabetic (OR=1.26), pre- diabetic (OR=1.11) and hypertensive (OR=1.28) individuals [65]. Analysis of 198,754 adults in the 2006 India DHS Survey found the population attributable fraction for TB due to low BMI decreased from 34.2% to 20.3% from the poorest to the richest wealth quintiles, while that of diabetes increased from 0.6% to 4.0% [66]. This finding may be attributable in part to an inverse relation between wealth and unhealthy food consumption (in the current study, we found an inverse relationship between GDQS- scores and asset quartile) and may suggest that the burden of TB attributable to diabetes will increase along with the nation's economic development.

To our knowledge, this is the first study to conduct detailed dietary or nutritional assessments jointly among TB cases (MDR or otherwise) and household and community controls (matched or otherwise). This allowed us to compare multiple facets of diet quality and nutrition status across strata of MDR-TB and nutritional risk in a population facing high burdens of both. However, care should be taken when generalizing results beyond high-burden slum areas of Mumbai. Analyses were powered for the primary objective of this study and not for this secondary analysis and hence we may have been underpowered to detect statistically significant differences between cases and controls with regards to certain nutrient-related variables – however, we observed a number of significant trends, suggesting reasonable statistical power. We did not have complete data characterizing cases including elements such as resistance patterns, which may have afforded a more comprehensive understanding of case status. However, this information likely did not materially impact on the focus and findings of the current analysis. We used two different types of control groups, household and community controls, both leading to similar conclusions regarding less optimal diets among cases. However, different types of control groups represent distinct sub-populations with their own advantages and disadvantages (for example, over-matching may be an issue in the case of family controls), and this was evident in the differences in diet between the household versus community controls [67]. Additionally, we conducted data collection during the COVID-19 pandemic, and although the FFQ focused on usual intake during the past year, we cannot rule out that more concurrent issues relating to food supply and diet may have influenced participants' responses. About one third of household and community controls had a positive IGRA test, indicating probable TB infection, which may have influenced their diet and nutritional status, and led to under-estimation of differences between cases and controls. Nonetheless we still observed strong differences in diet between cases and controls, reinforcing associations between diet and MDR-TB. Furthermore, our ability to infer causal relationships was limited by the cross-sectional design. Causality is supported by important potential mechanisms, but it is also plausible that diet quality and nutrition status deteriorated among cases following diagnosis due to impacts of the disease or clinical and financial impacts of treatment. While these impacts should be mitigated by restricting to cases who initiated treatment in the past month and who had no recent history of MDR-TB, disease could have affected diet if it remained undiagnosed or untreated for an extended period, which often occurs in India even in the presence of active case finding [68]. Recent diet may also disproportionately influence long-term recall [69], which could further limit our ability to capture diet prior to disease.

Thus, while this analysis supports compelling hypotheses, large prospective studies would be valuable to firmly establish the contribution of diet and nutrition status to MDR-TB infection and disease outcomes in India and other high-burden countries. A field-based, cluster-randomized controlled trial recently completed in Jharkhand (the RATIONS study) [70] has provided strong evidence for the benefit of nutritional support among both microbiologically-confirmed pulmonary DS-TB cases and household contacts. Among 10,345 contacts, provision of 6 months protein- and micronutrient-rich food supplements was associated with substantially reduced hazard of microbiologically-confirmed infection (aHR: 0.51, 95%CI

0.34-0.78) [71], 54.0% of supplemented cases (97.0% DS-TB) gained ≥5% of baseline weight gain at 2 months [49], and the latter was associated with substantially reduced hazard of death among cases (aHR: 0.39, 95% CI 0.18- 0.86). Given the comparative severity and duration of MDR-TB disease and treatment, the benefits of nutritional support on drug adherence, outcomes, recovery, and transmissibility among MDR-TB cases (and on prevention among household contacts) may be even more pronounced, and should be evaluated experimentally (drawing from the design of RATIONS). Integrating nutrition into MDR-TB programs, and indeed all TB programs, could therefore address key structural barriers and reduce the burden of disease.

Results of this study indicate an urgent need to improve nutrition in Mumbai slum dwellers – particularly among the MDR-TB-infected population, women, the elderly, and the poorest households – and highlight the potentially key role of nutrition interventions in reducing the burden of MDR-TB in urban India. Such interventions should focus on maintaining fruit and vegetable consumption, increasing that of lean protein sources, and decreasing that of refined grains, sweets and sugary beverages, and deep fried foods. Such interventions should also accompany strengthened TB and DR-TB control efforts [2,8] – including expanded drug supply and preventive treatment, active case detection, and management of comorbidities – and be placed in the context of broad multisectoral strategies [2,66,72,73] for poverty reduction, infrastructure and social development needed to effectively address multifactorial causes of TB in India (which will also sustain further improvements in nutrition).

## Supporting information

**S1 Table. Individual and household food consumption-related habits by population subgroup.**
(DOCX)

**S2 Table. Estimated marginal mean (95%CI) consumption of key food groups and nutrients, and diet metric scores by age category and asset quartile.**
(DOCX)

**S1 Checklist. Inclusivity in global research.**
(DOCX)

**S1 STROBE Checklist. An extension of the STROBE statement for nutritional epidemiology.**
(DOCX)

## Acknowledgments

The authors would like to thank Subhadra Mandalika, Fatima Kader, Mahtab Bamji, and Pratibha Dwarkanath for guidance in the design and interpretation of results of this analysis; Addanki Srivalli and Santhi Bhogadi for guidance in adapting methodology from the India Migration Study (IMS); and Mika Matsuzaki, Shilpa Bhupathiraju, and Sanjay Kinra for facilitating use of IMS materials. The authors gratefully acknowledge Lok Seva Sangam, a local non-governmental organization in Mumbai, for providing field workers who assisted in participant identification, household visits, and data collection.

## Author contributions

**Conceptualization:** Sabri Bromage, Yatin Dholakia, Wafaie W. Fawzi, Nerges Mistry.

**Data curation:** Laxmi Govekar, Sabri Bromage, Uttara Partap, Anupam Shukla, Prachi Dev.

**Formal analysis:** Laxmi Govekar, Sabri Bromage, Uttara Partap.

**Funding acquisition:** Wafaie W. Fawzi, Nerges Mistry.

**Investigation:** Laxmi Govekar, Anupam Shukla, Prachi Dev.

**Methodology:** Laxmi Govekar, Sabri Bromage, Uttara Partap, Yatin Dholakia, Wafaie W. Fawzi, Nerges Mistry.

**Project administration:** Yatin Dholakia, Wafaie W. Fawzi, Nerges Mistry.

**Resources:** Nerges Mistry.

**Software:** Sabri Bromage, Uttara Partap, Prachi Dev.

**Supervision:** Laxmi Govekar, Sabri Bromage, Anupam Shukla, Yatin Dholakia, Wafaie W. Fawzi, Nerges Mistry.

**Writing – original draft:** Laxmi Govekar, Sabri Bromage, Uttara Partap.

**Writing – review & editing:** Anupam Shukla, Prachi Dev, Yatin Dholakia, Wafaie W. Fawzi, Nerges Mistry.

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
