## [Decision Letter · Decision Letter 0]

21 Feb 2025

PGPH-D-24-02874

Diet and nutrition status of adult multidrug-resistant tuberculosis cases, household controls, and community controls in Mumbai, India

Dear Dr. Bromage,

Thank you for submitting your manuscript to PLOS Global Public Health. After careful consideration, we feel that it has merit but does not fully meet PLOS Global Public Health’s publication criteria as it currently stands. Therefore, we invite you to submit a revised version of the manuscript that addresses the points raised during the review process.

Specifically, the comments regarding the methodology section need to be addressed for further consideration.

We look forward to receiving your revised manuscript.

Kind regards,

Madhavi Bhargava

Academic Editor

Journal Requirements:

Additional Editor Comments (if provided):

Reviewers' comments:

Reviewer's Responses to Questions

**Comments to the Author**

1. Does this manuscript meet PLOS Global Public Health’s publication criteria?

Reviewer #1: Yes

Reviewer #2: No

Reviewer #3: Partly

2. Has the statistical analysis been performed appropriately and rigorously?

Reviewer #1: Yes

Reviewer #2: I don't know

Reviewer #3: No

3. Have the authors made all data underlying the findings in their manuscript fully available (please refer to the Data Availability Statement at the start of the manuscript PDF file)?

Reviewer #1: No

Reviewer #2: No

Reviewer #3: No

4. Is the manuscript presented in an intelligible fashion and written in standard English?

Reviewer #1: Yes

Reviewer #2: Yes

Reviewer #3: Yes

Reviewer #1: Introduction:

The introduction is overall well-written; however, for further improvement, it is recommended to:

Combine or merge the first and second paragraphs for better flow and coherence.

Begin by addressing the global burden of the problem and place the global context in the first paragraph to set the stage effectively.

Additionally, please ensure that findings from previous studies are reported accurately as presented in the original papers. For example, in lines 47 and 48 of the introduction, the author reported the risk of TB in urban residents as 1.56 (referencing paper no. 3). However, the original paper indicates that the risk is 1.58. Please revise this and carefully cross-check references throughout the manuscript.

Methodology:

The methodology section is somewhat unclear and lacks precision. It is recommended to:

Present the methods in a clearer and more structured manner.

Avoid jumping quickly from one point to another, as this may confuse readers and reduce their motivation to continue reading.

Results:

The results section would benefit from starting with a summary of the characteristics of the study participants. This should include key geographic, demographic, and socioeconomic details to provide the audience with a better understanding of the study population.

Tables:

To improve readability, it is recommended to link the discussion in the paragraphs with the corresponding tables. This will allow readers to easily navigate back and forth between the text and the tables.

Reviewer #2: Summary

The authors used data from a case-control study to explore the nutritional trends among MDR-TB cases in comparison to household and community controls. The authors report significant differences in nutritional status and dietary consumption between the cases and controls. I commend the authors for their undertaking of difficult nutritional assessments.

Major Issues:

Clarification and justification of methodological and statistical approaches is needed. The results section would benefit from streamlining and reorganization as presently comprehension is challenging.

Methods

Details are missing regarding timing and frequency of data collection.

Dietary assessment

Dietary assessment(s) is a fundamental part of this study, and as such each assessment should be sufficiently addressed. The authors mention the administration of a food frequency questionnaire; however, there is also mention of additional dietary assessments which are undefined. As currently written, it is unclear what nutritional data were collected, overall. There are minimal details on quantifying food consumption without stating how this data were initially collected.

Lines 134-136. The authors allude to food consumption data, but it is unclear where these data come from. Further, authors fail to provide justification for a 2000 kcal/day threshold. Explanation of “dish-bashed” food groups is necessary as readers may be unfamiliar with what this means substantively and nutritionally. It is also unclear whether these food items were converted into g/day and/or g/2000 kcal/day.

Lines 144-146. The inclusion of phytochemical index (PI) is questionable. The manuscript would benefit from elucidating the purpose PI serves nutritionally and in context to people living with MDR-TB. Presently, it is uncertain how this nutritional marker servers the aim of the paper and population of interest.

Lines 150-151. Authors state that food composition data was “supplemented as needed using data from the USDA and UK.” What was the justification for using two additional data sources and what data points were used from the USDA and UK, respectively?

Other assessments

Line 170. The manuscript states that household-food consumption related habits were collected, yet there is no indication of this assessment under dietary assessments and how these data were quantified. It is unclear whether the household measure is a distinct measure from the FFQ and other dietary assessments mentioned previously.

Statistical analysis

Line 194-195. Authors state that an asset score was computed and a principal component analysis employed. However, justification for why a PCA analysis was conducted is missing. Further, there is little information regarding the asset index. The methods section would benefit from describing what type of asset index was calculated and how this variable was operationalized.

Line 209-211. The manuscript details that regressions with interaction term(s) were conducted. It is unclear whether the authors are exploring a combined effect between variables or effect modification. Additionally, models include a diet metric that was energy-adjusted via residual method. The authors do not comment on the purpose for or what this adjustment achieves.

Results

Overall, the results were difficult to follow, and key points were not easily identified. This section was extensive and would benefit from being pared down. Data was expressed in large tables though figures would showcase the data nicely and improve interpretability.

Lines 323-333 Reporting beta coefficients is less interpretable than reporting relative measures.

Discussion

The authors did an insufficient job at exploring how nutritional support will reduce burden of MDR-TB, specifically. It seems, rather, that nutritional support will benefit those with TB infection regardless of drug susceptibility. Perhaps authors should explore how poor nutrition exacerbates disease outcomes and how potential interventions can mitigate adverse outcomes among MDR-TB patients.

The justification for community controls is underexplored and not well articulated. Discussion is lacking on how community data support the research conclusions and help advance the field.

This study was conducted during COVID-19 pandemic, and it is likely that the pandemic had direct impacts on food supply and nutritional intake. I would encourage the authors to address this, if possible.

Data availabilty

Data is accessible upon request. This does not meet the PLOS data policy.

Reviewer #3: This study is a commendable effort on describing in detail the nutritional intake of MDR-TB patients in comparison with their household contacts (HHC) and the community controls in the urban slum of Mumbai, where the burden of TB is high. However, the major issue is the screening of HHC only for symptoms of TB and not for the presence of TB infection (TBI) or sub-clinical TB. Presence of TBI or sub-clinical TB may play a role in their nutritional status. Other issue is regarding use analytical methods with multiple sub-groups with probably inadequate sample size for applying the statistical tests.

Overall the paper has many errors in spelling and language, which need to be corrected. Specific comments are as follows.

Introduction:

1. In the paragraph 1 lines 38-39, the number of cases of MDR-TB is mistakenly stated out of the proportion of infected.

2. The information on the proportion of untreated cases is not essential for this paper.

Methods:

1. Was the patient group homogenous? It will be good to mention the resistance pattern and the mode of diagnosis.

2. Standard terms should be used instead of terms such as non-tuberculotic household controls. HHC of TB patients might be having TB infection and therefore cannot be called non-tuberculotic.

3. Authors need to mention the validity of the method of estimating the quantity of food consumed by the use of Food Frequency Questionnaires. What is the percentage of error in the estimation of food consumption using this method?

4. Why the BMI classification for Asian Indians was not used?

5. Sample size estimation was not done. Was the sample size adequate for the multiple sub-group analysis (tertiles of age and quartiles of assets index as well as the nine groups stratified by sex)?

6. What was the rationale for categorising the age into <25, 25-40 and >40 years? How can the three parts of the tertiles be unequal?

7. Why the variables used in the multi-variable model were only limited to age, asset index, sex, and study arm? There are multiple confounders such as alcohol/ tobacco use and other co-morbidities, especially HIV, which have not been considered in the analysis.

Results: some of the results can be presented in graphs to reduce the lengthy texts.

Discussion: limitations need to be added.

**Do you want your identity to be public for this peer review?** For information about this choice, including consent withdrawal, please see our Privacy Policy

Reviewer #1: No

Reviewer #2: No

Reviewer #3: No

---

## [Editor Report · Decision Letter 1]

17 Dec 2025

Diet and nutrition status of adult multidrug-resistant tuberculosis cases, household controls, and community controls in Mumbai, India

PGPH-D-24-02874R1

Dear Dr. Bromage,

We are pleased to inform you that your manuscript 'Diet and nutrition status of adult multidrug-resistant tuberculosis cases, household controls, and community controls in Mumbai, India' has been provisionally accepted for publication in PLOS Global Public Health.

Best regards,

Academic Editor

Most comments by the reviewers have been addressed.

Kindly make a small correction in line 192, where the cut-offs mentioned are Asian, however the authors continue to write them as WHO cut-offs, kindly change that.

Moreover, the better reference will be the one that recommended separate cut-offs for Asian Indians: Misra A, Chowbey P, Makkar BM, Vikram NK, Wasir JS, Chadha D, et al; Consensus Group. Consensus statement for diagnosis of obesity, abdominal obesity and the metabolic syndrome for Asian Indians and recommendations for physical activity, medical and surgical management. J Assoc Physicians India. 2009 Feb;57:163-70.